

# Azhdarchid pterosaur diversity in the Bayanshiree Formation, Upper Cretaceous of the Gobi Desert, Mongolia

R. V. Pêgas[1], Xuanyu Zhou[2] and Yoshitsugu Kobayashi[3]

[1] Laboratório de Paleontologia, Serviço de Vertebrados, Museu de Zoologia da USP, São Paulo, São Paulo, Brazil
[2] College of Life Sciences, Shihezi University, Shihezi, Xinjiang Province, China
[3] Hokkaido University Museum, Hokkaido University, Sapporo, Hokkaido, Japan

Corresponding author
Xuanyu Zhou,
xyzhou@elms.hokudai.ac.jp

## ABSTRACT

Pterosaur remains are very rare in Mongolian Mesozoic deposits, in stark contrast with the great abundance of dinosaur fossils in the region. This contribution presents a reassessment of the azhdarchid pterosaur remains from the Bayanshiree Formation's "upper beds" (Turonian–Santonian), represented by two specimens coming from two distinct localities: the Burkhant and the Bayshin Tsav azhdarchids. These specimens, collected by the Japanese–Mongolian Joint Paleontological Expedition and originally described in 2009, have been previously interpreted as indeterminate azhdarchids. Under the light of current knowledge on the morphological diversity of azhdarchid cervical vertebrae, as well as on the taxonomic and phylogenetic signals these skeletal elements carry, we herein identify diagnostic features and reassess the phylogenetic affinities of the Bayanshiree azhdarchids in further detail. Our results suggest that the Burkhant azhdarchid, hereby named *Gobiazhdarcho tsogtbaatari* gen. et sp. nov., represents a medium-sized (3.0–3.5 meters in wingspan) basal member of a *Quetzalcoatlus-Arambourgiania* lineage. The Bayshin Tsav azhdarchid, *Tsogtopteryx mongoliensis* gen. et sp. nov., is recovered as a basal member of a *Hatzegopteryx*-lineage and, surprisingly, seems to represent a small form under 2 m in wingspan. Our results shed fresh light on the diversity and phylogeny of azhdarchid pterosaurs, and reinforce the reoccurring pattern of coexistence between multiple, differently-sized azhdarchid species from a same deposit.

## INTRODUCTION

Pterosaurs, the first vertebrate group to evolve powered flight, exhibit a fossil record stretching from the Late Triassic to the Cretaceous/Paleogene boundary, and an impressive diversity (*Wellnhofer, 1991*; *Witton, 2013*; *Jagielska & Brusatte, 2021*). Within pterosaurs, Azhdarchidae represents a very particular clade. Characterized mainly by their elongate cervical vertebrae with vestigial neural spines, azhdarchids are an almost ubiquitous presence in Turonian–Maastrichtian pterosaur assemblages worldwide, being the most

diverse and widespread group of pterosaurs during the Late Cretaceous (*Longrich et al., 2018*; *Andres, 2021*). Similar to other azhdarchoids, azhdarchids sported edentulous jaws and seem to have been relatively terrestrial in lifestyle compared to other pterosaurs (see *Witton & Naish, 2008*; *Witton & Habib, 2010*; *Witton, 2013*; but see *Averianov, 2014*). At present, the group includes at least 17 nominal species (see *Andres, 2021*; *Ortiz David, González Riga & Kellner, 2022*; *Zhou et al., 2024*; *Thomas et al., 2025*). Azhdarchids are well-known especially for including the largest flying creatures ever, comprising some gigantic forms with 10–11 meter-wingspans such as *Quetzalcoatlus northropi*, *Arambourgiania philadelphiae*, and *Hatzegopteryx thambema* (see *Witton & Habib, 2010*; *Andres & Langston, 2021*), as well as the ~9 meter-wingspan *Thanatosdrakon amaru* (*Ortiz David, González Riga & Kellner, 2022*). The group also includes smaller forms such as the 4.5 meter-wingspan *Quetzalcoatlus lawsoni* (*Andres & Langston, 2021*), the 3.5 meter-wingspan *Zhejiangopterus linhaiensis* (*Cai & Wei, 1994*), the 3.0 meter-wingspan *Eurazhdarcho langendorfensis* (*Vremir et al., 2013*), and, potentially, the ~1.6 meter-wingspan Hornby pterosaur (*Martin-Silverstone et al., 2016*).

Mesozoic deposits in Mongolia, especially in the Gobi Desert, are well-known for their rich fossil record (particularly regarding dinosaurs), resulting from an outstanding history of paleontological expeditions in the country (*e.g.*, *Andrews, 1932*; *Rozhdestvenskii, 1960*; *Kielan-Jaworowska, 1969*; *Lavas, 1993*; *Novacek, 1996*; *Colbert, 2000*; *Kurochkin & Barsbold, 2000*; *Watabe, Suzuki & Hayashibara Museum of Natural Sciences-Mongolian Paleontological Center Joint Paleontological Expedition, 2000*; *Watabe et al., 2010*). Still, in stark contrast to dinosaurs, the Mongolian record of pterosaurs is exceedingly scarce (*Watabe et al., 2009*; *Tsuihiji et al., 2017*). The most notable example is *Noripterus parvus*, which is so far the only nominal species of pterosaur from Mongolia (*Bakhurina, 1982*). This taxon is represented by several remains from the uppermost Jurassic–Lower Cretaceous Tsagan-Tsab Formation of Western Mongolia (see *Bakhurina, 1982*, *1986*; *Lü et al., 2009a*). The Mongolian pterosaur record only includes seven further specimens: an undescribed anurognathid from the Middle Jurassic beds of Bakhar in Central Mongolia (*Bakhurina & Unwin, 1995*), a tapejaroid cervical vertebra from the Lower Cretaceous Öösh Formation (*Andres & Norell, 2005*), an undescribed anhanguerid from the Albian Khuren Dukh Formation (*Bakhurina & Unwin, 1995*), two fragmentary azhdarchid specimens from the Upper Cretaceous Bayanshiree Formation (*Watabe et al., 2009*), a fragmentary ?azhdarchid long bone preserved as gut content of a dromaeosaurid specimen from the Upper Cretaceous Tugrikin Shireh beds (*Hone et al., 2012*), and fragmentary remains of a giant azhdarchid from the Maastrichtian Nemegt Formation (*Tsuihiji et al., 2017*). Only the last four findings come from the Gobi Desert (*Bakhurina & Unwin, 1995*; *Watabe et al., 2009*; *Tsuihiji et al., 2017*). Further pterosaur remains from the Gobi Desert include only some isolated bones from the Upper Cretaceous Iren Dabasu Formation in Inner Mongolia, China (*Currie & Eberth, 1993*), attesting the rarity of pterosaur remains in the Gobi region (perhaps linked to paleoecological factors; see *Averianov, 2014*).

The Bayanshiree azhdarchids comprise two specimens: the Bayshin Tsav azhdarchid, represented by an almost complete mid-cervical; and the Burkhant azhdarchid, comprising an atlantoaxis, a cervical III, and a partial mid-cervical (*Watabe et al., 2009*).

These specimens were collected by the *Hayashibara Museum of Natural Sciences-Mongolian Paleontological Center Joint Paleontological Expedition* in 1993 and 1995 from, respectively, the Bayshin Tsav and Burkhant localities (*Watabe et al., 2009*). These remains were originally described in detail by *Watabe et al. (2009)* and interpreted as indeterminate azhdarchids.

The present article aims at redescribing these specimens and investigating their phylogenetic relationships within Azhdarchidae, under the light of present-day data concerning azhdarchid diversity. Azhdarchid diversity and taxonomy have been the subject of many studies and great developments lately, and the taxonomic usefulness of cervical morphology has become increasingly evident for the group (*Vremir et al., 2015*; *Naish & Witton, 2017*; *Longrich et al., 2018*; *Pêgas et al., 2021*; *Andres, 2021*; *Andres & Langston, 2021*; *Zhou et al., 2024*).

## MATERIALS AND METHODS

### Geological Setting

The Bayanshiree Formation (also spelled Bayan Shireh, Baynshire, Bayshiree, or Baysheen Shireh Formation) is located in the eastern region of the Gobi Desert, in Mongolia. It can be divided in two members, informally dubbed as "upper" and "lower" beds (*Jerzykiewicz & Russell, 1991*; *Averianov & Sues, 2012*). The Bayanshiree Formation consists mostly of mudstones and sandstones, deposited in lacustrine to fluvial systems in a semi-arid paleoenvironment (*Shuvalov, 2000*; *Watabe et al., 2009*, *2010*).

The paleontological record of the Bayanshiree Formation is relatively rich, with a flourishing dinosaur fauna that includes ankylosaurids, pachycephalosaurids, ceratopsians, hadrosauroids, sauropods, and theropods (*e.g.*, *Benton et al., 2000*; *Watabe, Suzuki & Hayashibara Museum of Natural Sciences-Mongolian Paleontological Center Joint Paleontological Expedition, 2000*; *Watabe et al., 2010*; *Barsbold, Kobayashi & Kubota, 2007*; *Tsogtbaatar et al., 2019*). Notably, the turtle fauna is also very diverse, with numerous remains representing about eight species (*Danilov et al., 2014*). Mesozoic mammals (*Rougier, Davis & Novacek, 2015*; *Lopatin & Averianov, 2023*) and crocodylians (*Turner, 2015*) also occur. Pterosaur remains are very rare, and limited to the two specimens previously described by *Watabe et al. (2009)* and herein redescribed. As mentioned above, these pterosaur specimens come from outcrops of the Burkhant and Bayshin Tsav localities, both of which lie within the "upper beds" (*Averianov & Sues, 2012*).

Burkhant and Bayshin Tsav are well-known fossiliferous localities (Fig. 1), having been extensively explored in the literature before. The Burkhant locality (Dornogovi Province) is the same site that has yielded the holotype of the large dromaeosaurid theropod *Achillobator giganticus* (*Perle, Norell & Clark, 1999*), while the Bayshin Tsav locality (Ömnögovi Province) has yielded remains of the ornithomimosaur *Garudimimus brevipes* (*Kobayashi & Barsbold, 2005*), the therizinosaurids *Erlikosaurus andrewsi* and *Segnosaurus galbinensis* (*Perle, 1979*, *1981*; *Zanno, 2010*), the ankylosaurid *Talarurus plicatospineus* (*Park et al., 2020*), and the hadrosaurid *Gobihadros mongoliensis* (*Tsogtbaatar et al., 2019*).

The age of the Bayanshiree Formation has been the subject of several studies, and conclusions mostly converge towards a Cenomanian–Santonian age, summarized as

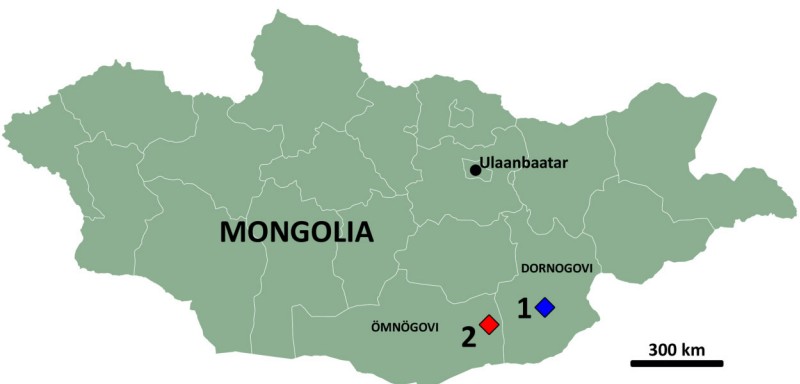

**Figure 1 Map of Mongolia.** Highlighted are the aimags (provinces) of Dornogovi (East Gobi) and Ömnögovi (South Gobi) and the localities of Burkhant (1) and Bayshin Tsav (2).

follows. Paleomagnetic analyses have indicated that the upper levels of the formation do not cross the Santonian–Campanian interval, meaning the upper limit of the Bayanshiree Formation is no younger than the latest Santonian (*Hicks et al., 1999*). Biostratigraphic analyses of the mollusk assemblages of the upper levels of the formation also suggest a Coniacian–Santonian upper limit (*Martinson, 1975*, *1982*). More recently, direct dating using calcite U-Pb measurements (taken from caliche samples) have indicated an age between 95.9 ± 6.0 and 89.6 ± 4.0 Ma, suggesting the Bayanshiree Formation lies somewhere within the Albian–Santonian interval (*Kurumada et al., 2020*). However, an age no older than the Cenomanian is indicated by both palynological and paleomagnetic evidence (*Hicks et al., 1999*; *Shuvalov, 2000*). Taken together, all of this data suggest that the Bayanshiree Formation is most likely Cenomanian–Santonian in age, as usually accepted (*Shuvalov, 2000*). Based on vertebrate faunal correlations, it has been suggested that the "lower beds" of the Bayanshiree Formation are Cenomanian–late Turonian in age, while the "upper beds" are late Turonian–Santonian in age (*Averianov & Sues, 2012*; *Danilov et al., 2014*).

## Anatomical terminology

The present work largely follows the anatomical terminologies provided by *Andres & Langston (2021)*. As such, we restrict the term 'preexapophysis' for the distinct process on the ventral surface of the prezygapophyseal pedicle (when present), as opposed to the 'preexapophyseal articulation' which is continuous with the cotylar surface (*Bennett, 2001*; *Andres & Langston, 2021*). Following *Andres & Langston (2021)*, we use the terms 'mid-cervical' for cervicals IV–VI, and 'middle-series cervicals' for cervicals III–VII. In this sense, we further suggest here the use of the term 'mesocervical' as a synonym for 'middle-series cervical', for brevity. We further adopt the terms 'accessory foramen' and 'adjacent foramen' for the pneumatic foramina located, respectively, dorsal and lateral to the neural canal (*Hone, Habib & Therrien, 2019*). Other anatomical terms are the same as in the work of *Andres & Langston (2021)*.

## Phylogenetic analysis

A phylogenetic analysis was performed in order to investigate the relationships of the new species herein analyzed within Azhdarchidae. For this purpose, we have utilized as a basis the phylogenetic matrix of *Zhou et al. (2024)*, with updates from *Manitkoon et al. (2025)* and *Thomas et al. (2025)*. As in *Pêgas (2024)*, the character list derives from a comprehensive survey of the literature (*Howse, 1986*; *Bennett, 1994*; *Kellner, 2003*, *2004*; *Unwin, 2003*; *Dalla Vecchia, 2009*, *2019*; *Lü et al., 2009b*; *Wang et al., 2012*; *Naish, Simpson & Dyke, 2013*; *Andres, Clark & Xu, 2014*; *Pêgas, Leal & Kellner, 2016*; *Pêgas, Costa & Kellner, 2018*; *Pêgas, Holgado & Leal, 2019*; *Pêgas et al., 2021*, *2023*; *Vidovic & Martill, 2018*; *Longrich et al., 2018*; *Holgado et al., 2019*; *Kellner et al., 2019*; *Zhou et al., 2019*, *2021*; *Andres, 2021*).

The analysis was conducted under maximum parsimony, using the software TNT 1.5 (*Goloboff & Catalano, 2016*). The analysis was divided in two steps, following the same protocol as previously described by *Wei et al. (2021)*. New Technology Search was used for the first step (using Sectorial Search, Ratchet, Drift and Tree fusing, default parameters), with random seed = 0. Subsequently, a Traditional Search swapping was performed using trees from RAM (using TBR, 10,000 replications, collapsing trees after search). All characters were treated with equal weights. A Mesquite file (Nexus format) containing the data matrix and a TNT file, ready for analysis execution in TNT, are available as Files S1 and S2, respectively.

As many of our characters/states pertain to cervical anatomy, care was taken to avoid characters affected by serial variation, taking the cervical series of *Q. lawsoni* and *W. brevirostris* as comparative bases. Azhdarchid species circumscriptions and coding sources are outlined below.

## Azhdarchid species circumscriptions

We follow *Vremir et al. (2015)* in considering that isolated, non-overlapping azhdarchid remains coming from a same deposit should be seen with caution, as the presence of multiple azhdarchid species in a same deposit seems to be a common, reoccurring condition. This can be seen, for example, in the Hațeg Basin (*Vremir et al., 2015*; *Solomon et al., 2020*), the Ouled Abdoun Basin (*Longrich et al., 2018*), and the Javelina Formation (*Andres & Langston, 2021*). For this reason, we regard that species circumscriptions for the taxa *Azhdarcho lancicollis*, *Aralazhdarcho bostobensis*, and *Cryodrakon boreas* require detailed revisions. Regarding *Azhdarcho lancicollis*, we note here that it autapomorphically exhibits a strongly sinusoidal medial margin of the prezygapophyseal peduncle, as can be seen in its type-specimen (*Averianov, 2010*). We herein provisionally restrict the circumscription of *Azhdarcho lancicollis* to specimens in which this same feature can be seen, such as ZIN PH 131/44 and ZIN PH 147/44 (see *Averianov, 2010*). Other referred specimens should be seen with caution, as Azhdarchidae indet. Similarly, the circumscription of *Cryodrakon boreas* is herein provisionally restricted to its holotype plus the two specimens that unambiguously exhibit some of its autapomorphic features, TMP 1993.40.11 and TMP 1989.36.254 (see *Hone, Habib & Therrien, 2019*), with other specimens being herein regarded as Azhdarchidae indet. Regarding *Aralazhdarcho*

*bostobensis*, its circumscription is herein provisionally restricted to its holotype (see *Averianov, 2007*). We further follow *Longrich et al., (2018)* for the circumscription of *Phosphatodraco mauritanicus*; *Andres & Langston (2021)* for the Javelina Fm. azhdarchids; *Vremir (2010)* and *Naish & Witton (2017)* for *Hatzegopteryx thambema*; and *Frey & Martill (1996)* and *Martill & Moser (2018)* for *Arambourgiania philadelphiae*.

## Phylogenetic nomenclature

The present work follows the PhyloCode (*de Queiroz & Cantino, 2020*) as a means of standardizing phylogenetic nomenclature. We primarily follow the phylogenetic definitions of *Andres (2021)* along with the updates of *Pêgas et al. (2021)* concerning azhdarchids. The phylogenetic nomenclatural scheme employed here, following recommendations of the PhyloCode (*de Queiroz, Cantino & Gauthier, 2020*), is presented in Table 1.

## Nomenclatural acts

The electronic version of this article in Portable Document Format (PDF) will represent a published work according to the International Commission on Zoological Nomenclature (ICZN), and hence the new names contained in the electronic version are effectively published under that Code from the electronic edition alone. This published work and the nomenclatural acts it contains have been registered in ZooBank, the online registration system for the ICZN. The ZooBank LSIDs (Life Science Identifiers) can be resolved and the associated information viewed through any standard web browser by appending the LSID to the prefix http://zoobank.org/. The LSID for this publication is urn:lsid:zoobank.org:pub:72240BB4-B98C-40B4-ADBC-A1DEDE9E06DA. The LSID for the new genus *Gobiazhdarcho* is: urn:lsid:zoobank.org:act:A675EB7D-3502-4F99-8446-8AE4918AC60A. The LSID for the species *Gobiazhdarcho tsogtbaatari* is: urn:lsid:zoobank.org:act:59F3DABA-6E84-4DE2-8948-F5F250E2E910. The LSID for the new genus *Tsogtopteryx* is: urn:lsid:zoobank.org:act:A724E4E3-A6EA-415E-9A12-0B8F0636FEF0. The LSID for the species *Tsogtopteryx mongoliensis* is: urn:lsid:zoobank.org:act:3F218EE5-2CB8-469D-A252-C5194C7C6911. The online version of this work is archived and available from the following digital repositories: PeerJ, PubMed Central SCIE and CLOCKSS.

## Materials availability

The two specimens herein redescribed are permanently stored at the Mongolian Paleontological Center (Mongolian Academy of Sciences, Ulaanbaatar, Mongolia), a public research institution and repository, where the specimens are available for research upon request. Specimen ID's are MPC–Nd 100/302 (the Burkhant azhdarchid; *Gobiazhdarcho tsogtbaatari* gen. et sp. nov.) and MPC–Nd 100/303 (the Bayshin Tsav azhdarchid; *Tsogtopteryx mongoliensis* gen. et sp. nov.). Furthermore, the dataset associated with our phylogenetic analysis is available as Files S1 (Nexus format) and S2 (TNT format).

**Table 1 Phylogenetic nomenclature.**

| Clade | Nominal and definitional authors, and Regnum code | Definition | Composition and remarks |
|---|---|---|---|
| Azhdarchoidea | *Unwin, 1995* [*Andres, 2021*], [355]. | Min ∇ (*Tapejara wellnhoferi* Kellner 1989 & *Quetzalcoatlus northropi Lawson, 1975*). | Includes the sister-taxa Tapejaromorpha and Azhdarchomorpha. |
| Azhdarchomorpha | *Pêgas et al., 2021* [*Pêgas et al., 2021*], [574]. | Max ∇ (*Azhdarcho lancicollis Nessov, 1984* ~ *Thalassodromeus sethi* Kellner & Campos 2002 & *Tapejara wellnhoferi* Kellner 1989). | Includes *Keresdrakon*, Chaoyangopteridae, and Azhdarchiformes. |
| Azhdarchiformes | *Andres, 2021* [*Andres, 2021*], [771]. | Max ∇ (*Quetzalcoatlus northropi Lawson, 1975* ~ *Chaoyangopterus zhangi* Wang & Zhou 2003). | Includes Alanqidae and Azhdarchidae. |
| Alanqidae | *Pêgas et al., 2021* [*Pêgas et al., 2021*], [576]. | Max ∇ (*Alanqa saharica Ibrahim et al., 2010* ~ *Chaoyangopterus zhangi* Wang & Zhou 2003 & *Azhdarcho lancicollis Nessov, 1984*). | Includes *Alanqa*, *Argentinadraco*, *Leptostomia*, and *Xericeps*. |
| Azhdarchidae (unrestricted emendation) | *Padian, 1986* {this work}, {1043} | Min ∇ (*Azhdarcho lancicollis Nessov, 1984*, *Phosphatodraco mauritanicus Suberbiola et al., 2003*, *Zhejiangopterus linhaiensis Cai & Wei, 1994*, & *Quetzalcoatlus northropi Lawson, 1975*). | See main text for a detailed protologue including remarks on its composition and conceptualization. |
| Phosphatodraconia (new clade name) | This work [this work], [1044]. | Max ∇ (*Phosphatodraco mauritanicus Suberbiola et al., 2003* ~ *Azhdarcho lancicollis Nessov, 1984*, *Zhejiangopterus linhaiensis Cai & Wei, 1994* & *Quetzalcoatlus northropi Lawson, 1975*). | Includes *Aralazhdarcho*, *Eurazhdarcho*, *Phosphatodraco*, and *Wellnhopterus*. |
| Quetzalcoatlida (new clade name) | This work [this work], [1045]. | ∇ apo piriform/clithridiate neural canal opening [*Quetzalcoatlus lawsoni Andres & Langston, 2021*]. | Includes Hatzegopterygia and Quetzalcoatlini. |
| Hatzegopterygia (new clade name) | This work [this work], [1046]. | Max ∇ (*Hatzegopteryx thambema Buffetaut, Grigorescu & Csiki, 2002* ~ *Quetzalcoatlus northropi Lawson, 1975*, *Azhdarcho lancicollis Nessov, 1984*). | Includes *Albadraco*, *Cryodrakon*, *Hatzegopteryx*, the Pui azhdarchid, and *Tsogtopteryx*. |
| Quetzalcoatlini (new clade name) | This work [this work], [1047]. | Max ∇ (*Quetzalcoatlus northropi Lawson, 1975* ~ *Hatzegopteryx thambema Buffetaut, Grigorescu & Csiki, 2002*, *Azhdarcho lancicollis Nessov, 1984*). | Includes *Arambourgiania*, *Gobiazhdarcho*, *Infernodrakon*, *Nipponopterus*, *Quetzalcoatlus*, and *Thanatosdrakon*. |

**Note:**
Reference phylogeny: this work. Original definitional authors and Regnum codes are given between square brackets. Authors and Regnum codes of unrestricted emended definitions are given between curly braces. See the main text for further comments on the diagnoses, compositions, and conceptualizations of azhdarchid clades.

# SYSTEMATIC PALEONTOLOGY

Pterosauria *Owen, 1842 sensu* [*Andres & Padian, 2020a*]

Pterodactyloidea *Plieninger, 1901 sensu* [*Andres & Padian, 2020b*]

Azhdarchoidea *Unwin, 1995 sensu* [*Andres, 2021*]

**Azhdarchidae** *Padian, 1986 sensu* {this work}

**Definition (unrestricted emendation).** The least inclusive clade containing *Azhdarcho lancicollis Nessov, 1984*, *Phosphatodraco mauritanicus Suberbiola et al., 2003*, *Zhejiangopterus linhaiensis Cai & Wei, 1994*, and *Quetzalcoatlus northropi Lawson, 1975*. RegNum registration number: {1,043}. This is an unrestricted emendation (original PhyloCode-based definition: *Andres, 2021*).

**Composition.** *Aerotitan sudamericanus*, *Aralazhdarcho bostobensis*, *Azhdarcho lancicollis*, *Eurazhdarcho langendorfensis*, *Mistralazhdarcho maggi*, *Phosphatodraco mauritanicus*, *Wellnhopterus brevirostris*, *Zhejiangopterus linhaiensis*, and Quetzalcoatlida (see below).

**Diagnostic synapomorphies.** Mid-cervical vertebrae neural spines vestigial (composed of anterior and posterior neural processes, connected by a neural ridge); mid-cervical vertebrae extremely elongated (maximum length/width ratio over 5); wing digit phalanges 2 and 3 bearing a ventral keel.

**Remarks.** Azhdarchids have traditionally and consistently been diagnosed by their unique cervical osteology, most notably characterized by extremely elongated vertebrae with vestigial neural spines (*Padian, 1986*; *Kellner, 2003*; *Unwin, 2003*; *Andres, 2021*). When originally defined in accordance with the PhyloCode, this clade reflected such traditional and consistent usage, at least under the context of the original definitional reference phylogeny (*Andres, 2021*). Azhdarchidae was then defined as the least inclusive clade containing *Azhdarcho lancicollis* and *Quetzalcoatlus northropi* (*Andres, 2021*). In this sense, azhdarchids *sensu Andres (2021)* share the following apomorphies: middle-series cervical vertebrae maximum length/width ratio over 4.8; middle-series cervical neural spines vestigial; and wing digit phalanges 2 and 3 bearing a ventral keel; and comprise the same nominal species as in here (with the addition of *Thanatosdrakon amaru*, *Nipponopterus mifunensis* and *Infernodrakon hastacollis* here, as well as the herein named taxa).

As regulated by the PhyloCode (*de Queiroz & Cantino, 2020*), an unrestricted emendation is "*a mechanism to prevent undesirable changes in the application of a particular name (in terms of clade conceptualization) when the original definition is applied in the context of a revised phylogeny*" (Article 15.11). This is the case here, where direct application of the original definition would leave a large number of taxa consistently and universally included in Azhdarchidae outside of the group; thereby severely disrupting not only the composition of Azhdarchidae but also its diagnosis. Therefore, fulfilling Article 15 of the PhyloCode and aiming at preserving the clade's original diagnosis and composition under the context of the present phylogeny, an unrestricted emendation is herein proposed. It is worth noting that, for workers who may prefer alternative phylogenies, the previous definition (*Andres, 2021*) will still take precedence, following Article 15.15 of the PhyloCode (application of the present emendation under alternative phylogenies would not affect clade conceptualization).

**Quetzalcoatlida** new clade name (Table 1)

**Quetzalcoatlini** new clade name (Table 1)

***Gobiazhdarcho tsogtbaatari*** gen. et sp. nov.

**Holotype.** MPC–Nd 100/302 (Figs. 2–4), the Burkhant azhdarchid (*Watabe et al., 2009*). The specimen includes the atlantoaxis, cervical III, and fragmentary cervical VI (*Andres & Langston, 2021*).

**Etymology.** The generic epithet is a combination of the words *Gobi*, in reference to the Gobi Desert, and *azhdarcho* (from Persian *azhdar*, a dragon-like creature), a common suffix for azhdarchid pterosaurs. The specific epithet honors Khishigjav Tsogtbaatar, in recognition of his contributions to Mongolian vertebrate paleontology.

**Type locality and horizon.** Bluish white siltstone layer, Burkhant locality, Eastern Gobi Aimag (*Watabe et al., 2009*). Upper Bayanshiree Formation, Turonian–Santonian (see *Averianov & Sues, 2012*).

**Diagnosis**. The new azhdarchid taxon exhibits the following combination of features regarding mesocervical vertebrae morphology (including five autapomorphies, marked with an asterisk): prezygapophyseal pedicle with a large ventral tubercle anterior to the preexapophyses*; postexapophyses reduced (barely extending posterior to the condyle; shared with *Nipponopterus mifunensis*); presence of an interpostexapophyseal lamina*; presence of five longitudinal ridges on the ventral surface of the postexapophyses*; presence of epipophyseal keels (shared with *Nipponopterus mifunensis*); epipophyseal keels bearing an acuminate apex*; CVI epipophyses strongly curving medially*.

**Description and comparisons.** Even though this specimen has already been explored in detail by *Watabe et al. (2009)*, a redescription with complementary details and comparisons is herein presented. The specimen includes three elements: a complete atlantoaxis, a roughly complete cervical III, and a fragmentary cervical VI.

**Atlantoaxis.** The atlas and axis are fused into an atlantoaxis (Fig. 2), as is typical in relatively mature ornithocheiroids (*e.g.*, *Andres, 2021*). The fusion is complete, and no clear traces of sutures could be found. The element is roughly complete, except for some minor damage on the dorsal edge of the neural spine.

Delimitations between atlanteal elements cannot be discerned. The cotyle of the atlas is subcircular in shape (Figs. 2A, 2D), unlike the piriform (ventrally acuminate) shape found in cf. *Azhdarcho lancicollis* (ZIN PH 105/44; *Averianov, 2010*) and *Quetzalcoatlus lawsoni* (*Andres & Langston, 2021*), or the subelliptical shape seen in the indeterminate azhdarchid ZIN PH 54/43 (tentatively attributed to *Aralazhdarcho bostobensis*; *Averianov, 2007*). Above the cotyle, the neural canal is as (laterolaterally) wide as, and (dorsoventrally) lower than, the cotyle, as in both cf. *Azhdarcho lancicollis* (*Averianov, 2010*) and *Quetzalcoatlus lawsoni* (*Andres & Langston, 2021*). Still, it differs from both of these forms in that the neural canal opening is relatively more dorsoventrally depressed (roughly semicircular), rather than subtriangular. It also differs from ZIN PH 54/43, in which the neural canal is subcircular and less wide than the cotyle (*Averianov, 2007*). Ventral to the cotyle, a gentle midline process projecting anteriorly is present (presumably the intercentrum; Fig. 2K); much more discrete than in either cf. *Azhdarcho lancicollis* (*Averianov, 2010*) or *Quetzalcoatlus lawsoni* (*Andres & Langston, 2021*). The lateral surface of the atlas is pierced by a distinctive intervertebral foramen, as in cf. *Azhdarcho lancicollis* (*Averianov, 2010*) and *Quetzalcoatlus lawsoni* (*Andres & Langston, 2021*). In the Burkhant azhdarchid,

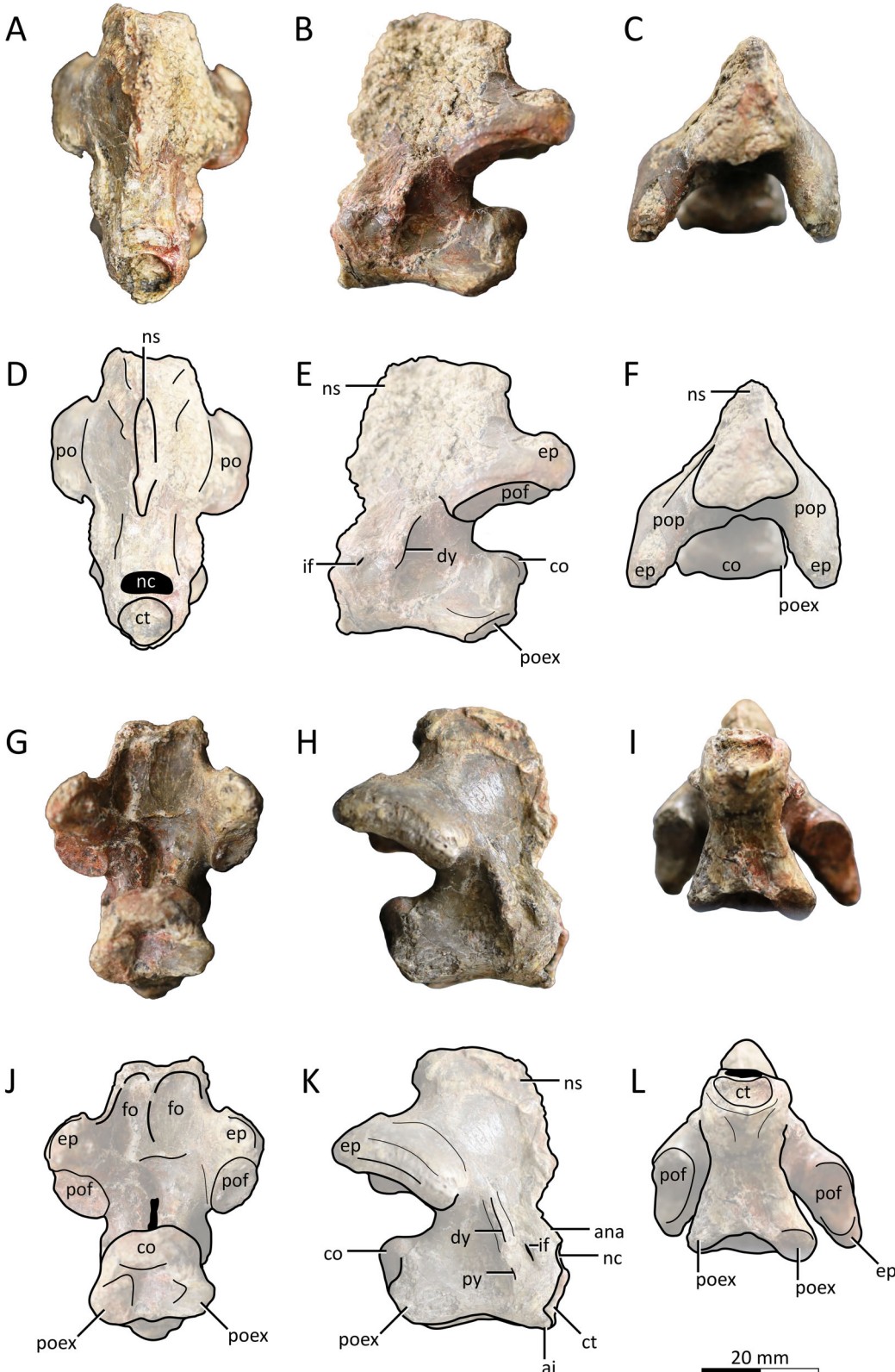

**Figure 2 MPC–Nd 100/302a, atlantoaxis of the holotype of *Gobiazhdarcho tsogtbaatari*.**
(A) Anterodorsal view; (B) left lateral view; (C) dorsal view; and (D–F) respective schematic drawings.

**Figure 2** (continued)
(G) Posterior view; (H) right lateral view; and (I) ventral view; and (J–L) schematic drawings. Abbreviations: ana, atlas neural arch; ai, atlas intercentrum; co, condyle; ct, cotyle; dy, diapophysis; ep, epipohysis; fo, fossa; if, intervertebral foramen; nc, neural canal; ns, neural spine; po, postzygapophysis; poex, postexapophysis; pof, postzygapophyseal facet; pop, postzygapophyseal pedicle; py, parapophysis. Scale bar = 20 mm.

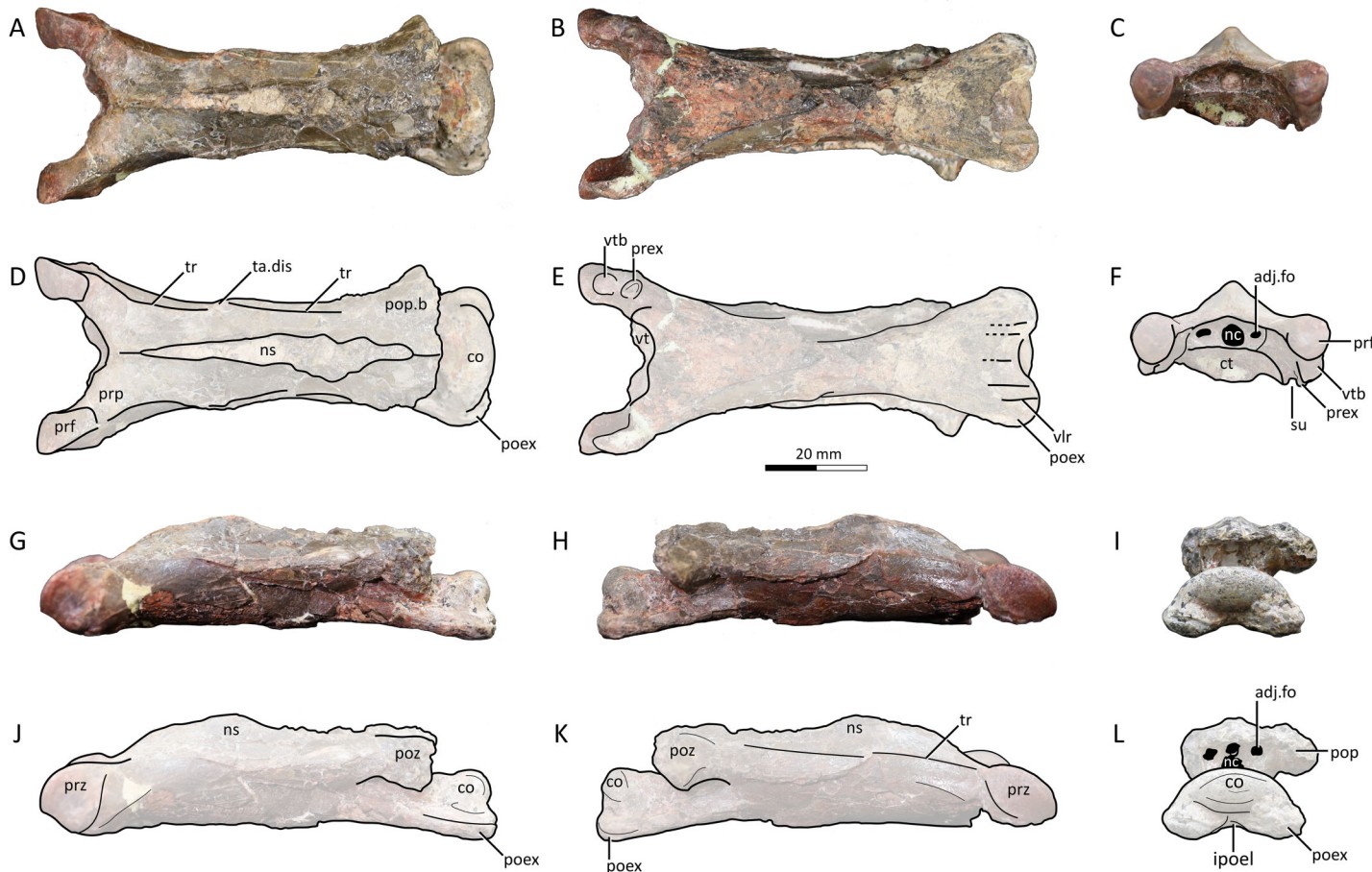

**Figure 3** **MPC–Nd 100/302b, cervical III of the holotype of *Gobiazhdarcho tsogtbaatari*.** (A) Dorsal view; (B) ventral view; (C) anterior view; and (D–F) respective schematic drawings. (G) Left lateral view; (H) right lateral view; and (I) posterior view; and (J–L) schematic drawings. Abbreviations: adj.fo, adjacent foramen; co, condyle; ct, cotyle; dy, diapophysis; ipoel, interpostexapophyseal ridge; nc, neural canal; ns, neural spine; poex, postexapophysis; pop, postzygapophyseal pedicle; pop.b, postzygapophyseal pedicle base; poz, postzygapophysis; prex, preexapophysis; prf, prezygapophyseal facet; prp, prezygapophyseal pedicle; prz, prezygapohysis; su, sulcus; ta.dis, taphonomic distortion; tr, transverse ridge; vlr, ventral longitudinal ridge; vtb, ventral tubercle. Scale bar = 20 mm.

this foramen lies ventral to the level of the axis condyle, similar to *Quetzalcoatlus lawsoni* but unlike cf. *Azhdarcho lancicollis*, in which it lies further higher (*Averianov, 2010*).

The axis lacks any pneumatic foramina (Fig. 2), similar to cf. *Azhdarcho lancicollis* and *Quetzalcoatlus lawsoni* but unlike ZIN PH 54/43 and *Mistralazhdarcho maggi* in which a lateral pneumatic foramen is present (*Averianov, 2007*; *Vullo et al., 2018*). The neural spine

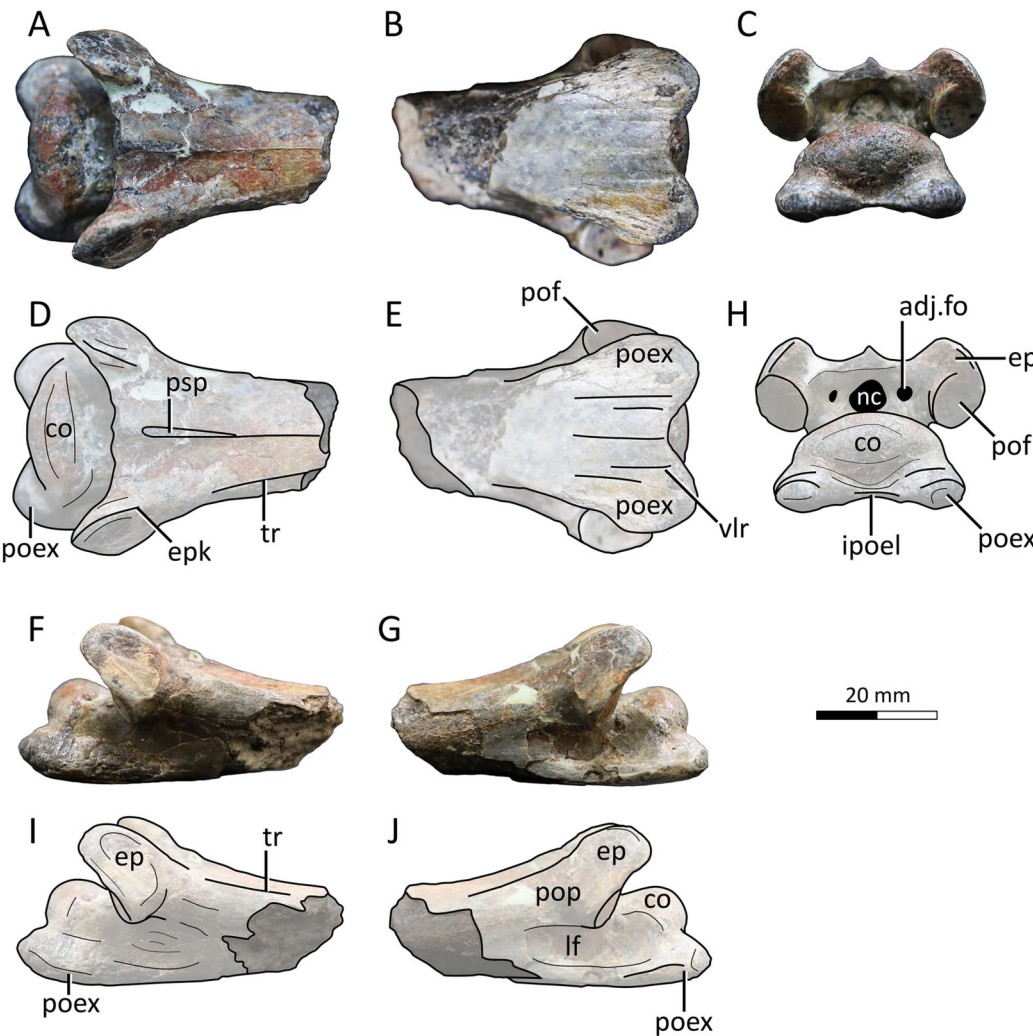

**Figure 4 MPC–Nd 100/302c, cervical VI of the holotype of *Gobiazhdarcho tsogtbaatari*.** (A) Dorsal view; (B) ventral view; (C) posterior view; and (D–F) respective schematic drawings. (F) Right lateral view; (H) left lateral view; and (I, J) schematic drawings. Abbreviations: adj.fo, adjacent foramen; co, condyle; ep, epipophyis; epk, epipophyseal keel; ipoel, interpostexapophyseal ridge; lf, lateral fossa; nc, neural canal; poex, postexapophysis; pof, postzygapophyseal facet; pop, postzygapophyseal pedicle; psp, posterior spinous process; tr, transverse ridge; vlr, ventral longitudinal ridge. Scale bar = 20 mm.

is reduced (occupying only the first half of the atlantoaxis length) and anteriorly-facing, thus being excluded from the posterodorsal apex of the element (Figs. 2B, 2C, 2H). This is similar to the condition seen in *Quetzalcoatlus lawsoni* (*Andres & Langston, 2021*) though less reduced, and quite distinct from the more typical condition seen in cf. *Azhdarcho lancicollis*, wherein the neural spine forms the posterodorsal apex of the element (*Averianov, 2010*). The posterior face of the neural arch, above the postzygapophyseal pedicles, bears a pair of well-developed fossae. This feature is absent in cf. *Azhdarcho lancicollis* and *Quetzalcoatlus lawsoni*. The postzygapophyseal pedicles are massively constructed, roughly as dorsoventrally tall as the centrum. The epipophyses are

well-developed, as in cf. *Azhdarcho lancicollis* (*Averianov, 2010*) and *Quetzalcoatlus lawsoni* (*Andres & Langston, 2021*), but differ from these forms in being dorsally keeled rather than blunt. The posterior neural canal opening is dorsoventrally elongate and slit-like in shape (Figs. 2G, 2J), unlike the subcircular conditions seen in cf. *Azhdarcho lancicollis*, *Quetzalcoatlus lawsoni*, and *Mistralazhdarcho maggi*. The condyle is gently cordate (Figs. 2G, 2J), similar to ZIN PH 54/43 (*Averianov, 2007*) and cf. *Azhdarcho lancicollis* (*Averianov, 2010*) but unlike the ovoid shape found in *Quetzalcoatlus lawsoni* (*Andres & Langston, 2021*) and *Mistralazhdarcho maggi* (*Vullo et al., 2018*).

**Cervical III.** The third cervical is almost completely preserved (Fig. 3), despite damage on the neural spine and postzygapophyses, the latter of which are mostly missing. It is moderately elongated, with a maximum length/width ratio of 4.4; lower than the same ratio in *Q. lawsoni* (7.4; *Andres & Langston, 2021*) but similar to *Eurazhdarcho langendorfensis* (~4) and ?*A. lancicollis* (4.38; *Averianov, 2010*), and slightly higher than in *Phosphatodraco mauritanicus* (~3; *Suberbiola et al., 2003*; *Kellner, 2010*) and *Zhejiangopterus linhaiensis* (~3; *Cai & Wei, 1994*). The neural spine is tall, as in all other known azhdarchid third cervicals (*Averianov, 2010*; *Vremir et al., 2013*; *Andres & Langston, 2021*). The neurocentral shaft shows taphonomic distortion caused by an expansion (presumably caused by mineral infiltration or expansion) at the anterior third of its length (Figs. 3A, 3D). A similar distortion can be seen in *Quetzalcoatlus lawsoni* specimen TMM 41544-15 (*Andres & Langston, 2021*). Disregarding such distortion, the lateral margins of the vertebra seem to have been gently curved in ventral and dorsal views.

The neural spine is single (not bifid), as typical of azhdarchid third cervicals and unlike azhdarchid mid-cervicals (*Andres & Langston, 2021*). It extends for almost the entire dorsal surface of the neural arch (Figs. 3A, 3D), as in *Eurazhdarcho langendorfensis* (*Vremir et al., 2013*) and cf. *Azhdarcho lancicollis* (ZIN PH 131/44; *Averianov, 2010*) but unlike *Quetzalcoatlus lawsoni* in which it is more restricted towards the mid-shaft (*Andres & Langston, 2021*). On the anterior vestibule, it can be seen that the neural canal opening is piriform (Fig. 3C), being mostly subcircular except for a tapered dorsal apex. A pair of adjacent foramina are present. The adjacent foramina are approximately aligned with the center of the neural canal opening. The prezygapophyseal pedicles bear a ventral tubercle just anterior to the preexapophyses on both sides (Figs. 3C, 3F). The prezygapophyseal articular facets are anteriorly expanded with a squared-off anterior margin, unlike the elliptical shape seen in cf. *Azhdarcho lancicollis* (*Averianov, 2010*) and *Quetzalcoatlus lawsoni* (*Andres & Langston, 2021*).

The vertebra exhibits a transverse ridge that is slightly dorsally reflected (Figs. 3A, 3D), similar to mid-cervicals of *Quetzalcoatlus lawsoni* and *Arambourgiania philadelphiae* (*Andres & Langston, 2021*). Ventral to this dorsolateral ridge lies a lateral ridge, which extends from the lateral surface of the base of the prezygapophyseal pedicle until the mid-length of the vertebra. As previously noted (*Andres & Langston, 2021*), the ventral surface produces a ventral tumescence on the anterior region (posterior to a small, blunt hypapophysis) extending posteriorly as a raised band until the mid-length of the vertebrae, giving the anterior half of the ventral margin a convex aspect in lateral view, similar to the

third cervicals of *Eurazhdarcho langendorfensis* (*Vremir et al., 2013*), cf. *Azhdarcho lancicollis* (*Averianov, 2010*), and *Quetzalcoatlus lawsoni* (*Andres & Langston, 2021*).

The posterior vestibule houses a clithridiate neural canal (Fig. 3I), which exhibits an incipient (?vestigial) bony bar that almost separates it into neural canal and accessory foramen. A pair of adjacent foramina are present. The postzygapophyses are almost entirely lacking, and so not much can be observed. The condyle is ovoid in shape, with a convex dorsal margin in posterior view. This is similar to the condition seen in ?*Azhdarcho lancicollis* (*Averianov, 2010*), but unlike the chordate condition seen in *Quetzalcoatlus lawsoni* (*Andres & Langston, 2021*). The postexapophyses are quite anteroposteriorly short. The ventral surface of the centrum between the postexapophyses produces a transverse lamina, which connects the two postexapophyses. The ventral surface between the postexapophyses is concave, though not developed into a fossa. Despite superficial damage, it can be seen that a set of five longitudinal ridges are present on this ventral surface (two paired ones plus a median one). A single, median ridge is located on the sagittal line, and a further pair of ridges is located on each postexapophysis (Fig. 3E). This represents an autapomorphy of *Gobiazhdarcho tsogtbaatari*, and the same feature can be seen in the cervical VI (see below).

**Cervical VI.** The third element, MPC–Nd 100/302c, comprises the posterior region of a mesocervical (Fig. 4). It has been originally interpreted as an indeterminate mesocervical (*Watabe et al., 2009*), and later reinterpreted as either a cervical IV (*Averianov, 2014*) or a cervical VI (*Andres & Langston, 2021*). The latter interpretation is corroborated here, as explored further below.

The postzygapophyseal pedicles exhibit robust epipophyses, as typical of azhdarchid cervicals (*e.g.*, *Averianov, 2010*; *Andres & Langston, 2021*). The long axis of each epipophysis curves medially, similar to the sixth cervical of *Quetzalcoatlus lawsoni* (*Andres & Langston, 2021*). Still, as seen from the posterior view, the outer edge of each epipophysis is strongly curved medially, much more markedly than in *Quetzalcoatlus lawsoni* or any other azhdarchid (Fig. 4). Furthermore, the anterodorsal surface of each epipophysis is distinctively keeled (Figs. 4D, 4H), similar to the condition seen in the atlantoaxis (suggesting this feature may have been general for the cervical series in this taxon). This differs from the usual condition seen in other azhdarchids wherein the dorsal epipophyseal surface is blunt, as seen in *Quetzalcoatlus lawsoni* (all mesocervicals; *Andres & Langston, 2021*), *Cryodrakon boreas* (*Hone, Habib & Therrien, 2019*), the Bissekty azhdarchid material referred to *Azhdarcho lancicollis* (all mesocervicals; *Averianov, 2010*), *Quetzalcoatlus* cf. *northropi* (*Andres & Langston, 2021*), *Albadraco tharmisensis* (*Solomon et al., 2020*), and the Bakony azhdarchid MTMGyn/450 (*Ősi & Weishampel, 2005*). Similarly, *Nipponopterus mifunensis* also exhibits epipophyses with a sharp dorsal surface (*Ikegami, Kellner & Tomida, 2000*; *Zhou et al., 2024*). However, this condition is slightly distinct in this form. In *Nipponopterus mifunensis*, CVI bears a pair of elevated, distinct keels (with concave edges in cross-section) that extend throughout the entire dorsal surface of the postzygapophyseal peduncles, including the epipophyses (*Zhou et al., 2024*). In contrast, *Gobiazhdarcho tsogtbaatari* exhibits epipophyseal keels that are restricted to the

epipophyses and acuminate in shape (with convex edges). The shape of the epipophyseal keel of *Gobiazhdarcho tsogtbaatari* is therefore considered as autapomorphic.

The dorsal surface exhibits dorsally reflected transverse ridges (Figs. 4A, 4D), similar to MPC–Nd 100/302b as well as *Quetzalcoatlus lawsoni*, *Arambourgiania philadelphiae*, and *Nipponopterus mifunensis* (*Andres & Langston, 2021*; *Zhou et al., 2024*). The interpostzygapophyseal lamina is sinusoidal (in dorsal view), with a convex mid-region, similar to cervicals VI and VII of *Quetzalcoatlus lawsoni* but unlike cervical V in which this lamina is concave (*Andres & Langston, 2021*). The posterior neural opening is higher than wide and is piriform in shape, with an acuminate apex. The neural canal is bordered by a small pair of adjacent pneumatic foramina, which are aligned with the center of the neural canal opening. There is no individualized accessory pneumatic foramen dorsal to the neural canal.

The condyle is ovoid in shape, with a prominently convex dorsal margin in posterior view, similar to cervical VI (and unlike the cordate shape seen in other mesocervicals) of *Quetzalcoatlus lawsoni* (*Andres & Langston, 2021*). A pair of postexapophyses are present lateroventrally to the condyle, as typical of ornithocheiroids as well as of ctenochasmatids (*Andres & Ji, 2008*; *Andres, 2021*). Similar to cervical III, the postexapophyses of cervical VI are relatively reduced in anteroposterior length, barely extending the posterior limits of the condyle when seen in dorsal view (Figs. 4A, 4D). This contrasts with the mesocervicals of other azhdarchids, especially considering that in the complete cervical series of *Quetzalcoatlus lawsoni* the postexapophyses of CVI are the longest ones (*Andres & Langston, 2021*). On the posterior surface of the centrum, a well-developed horizontal lamina connects the two postexapophyses, again similar to cervical III (Figs. 4C, 4H). This feature, hereby termed an interpostexapophyseal lamina, is unique to *Gobiazhdarcho tsogtbaatari*, being absent in any other known azhdarchid cervical. As in the third cervical (see above), the ventral surface of the postexapophyses bears a set of five ventral parasagittal ridges, organized as follows. Each postexapophysis exhibits two ventral parasagittal ridges, and a single ventral parasagittal ridge is further present between the two postexapophyses (Figs. 4B, 4E). This feature is also unique to *Gobiazhdarcho tsogtbaatari*.

We corroborate here the interpretation of *Andres & Langston (2021)*, identifying element MPC–Nd 100/302c as a cervical VI. Most notably, the posterior condyle is ovoid in shape when seen in posterior view, with a significantly convex dorsal margin. If compared to the cervical series of *Quetzalcoatlus lawsoni*, this is most similar to cervical VI, and distinct from the rather cordate shape seen in cervicals IV, V, and VII (*Andres & Langston, 2021*). Furthermore, we consider MPC–Nd 100/302c unlikely to represent a cervical IV due to the lack of a posterior deflection of the centrum (*Andres & Langston, 2021*).

**Ontogenetic assessment.** Inferring ontogenetic stages in pterosaurs is a controversial task subject (*Bennett, 1993*; *Kellner, 2015*; *Dalla Vecchia, 2018*). Based on current evidence, it is clear that, during pterosaur ontogeny, the fusion between neural arches and centra is asynchronous throughout the vertebral column, and it is also clear that this occurs early in ontogeny concerning cervical vertebrae (*e.g.*, *Eck, Elgin & Frey, 2011*; *Shen et al., 2021*). In
contrast, fusion between atlas and axis (forming the atlantoaxis complex) seems restricted to individuals that are close to osteological maturity (*e.g.*, *Kellner, 2015*). Therefore, full atlantoaxis fusion in MPC–Nd 100/302 indicates that it is close to osteological maturity. Furthermore, specimen MPC–Nd 100/302 also exhibits a fairly smooth and dense bone surface, which is typical of adult pterosaur bones (*e.g.*, *Bennett, 1993*), including cervicals (*Vremir et al., 2015*; *Longrich et al., 2018*; *Hone, Habib & Therrien, 2019*; *Smith, Martill & Zouhri, 2023*). This is unlike the "grained texture" found in young pterosaur bones (*e.g.*, *Bennett, 1993*), including cervicals (*Hone, Habib & Therrien, 2019*; *Solomon et al., 2020*). On the other hand, the third cervical of MPC–Nd 100/302 lacks fused ribs. This suggests that MPC–Nd 100/302 was close to, but had not yet reached, full osteological maturity, and may be regarded as a late subadult.

**Hatzegopterygia** new clade name (Table 1)

*Tsogtopteryx mongoliensis* gen. et sp. nov.

**Holotype.** MPC–Nd 100/303, the Bayshin Tsav azhdarchid (*Watabe et al., 2009*). The specimen comprises an almost complete cervical VI (Fig. 5).

**Etymology.** The generic epithet is a combination of the words *Tsogt* (as in Tsogtbaatar; Mongolian: *mighty hero*) and *pteryx* (Ancient Greek: *wing*). The specific epithet refers to the provenance of the type specimen.

**Type locality and horizon.** Northwestern part of the Bayshin Tsav locality, Southern Gobi Aimag (*Watabe et al., 2009*). Upper Bayanshiree Formation, Turonian–Santonian (see *Averianov & Sues, 2012*).

**Diagnosis.** The new azhdarchid taxon exhibits the following combination of features regarding mesocervical vertebrae morphology (including two autapomorphies, marked with an asterisk): transverse ridges lateralized (shared with other non-quetzalcoatlinins); CVI prezygapophyseal pedicle bearing a sharp ventral keel*; CVI centrum bearing a pair of longitudinal ventrolateral ridges*; anterior neural spine process larger than the posterior one (shared with other hatzegopterygians).

**Description and comparisons.** MPC–Nd 100/303 is an elongated mesocervical vertebra. It is almost completely preserved, except for the posterior region. The element is broken posteriorly at about the level of the base of the postzygapophyseal pedicles, so that the postzygapophyses, posterior vestibule, condyle, and postexapophyses are missing. The neural spine is bifid, divided into anterior and posterior processes. The anterior process is conspicuously larger than the posterior one, as in *Albadraco tharmisensis* (*Solomon et al., 2020*), the Pui azhdarchid (*Vremir et al., 2015*), *Cryodrakon boreas* (*Hone, Habib & Therrien, 2019*), and, to a lesser extent, cf. *Hatzegopteryx thambema* (*Vremir, 2010*). In dorsal view, the lateral margins of the centrum are gently curved, but the vertebral shaft does not exhibit a strong constriction. The transverse ridges are positioned laterally (Fig. 5), as in *Azhdarcho lancicollis* (*Averianov, 2010*), *Cryodrakon boreas* (*Hone, Habib & Therrien, 2019*), and *Phosphatodraco mauritanicus* (*Longrich et al., 2018*), and unlike the

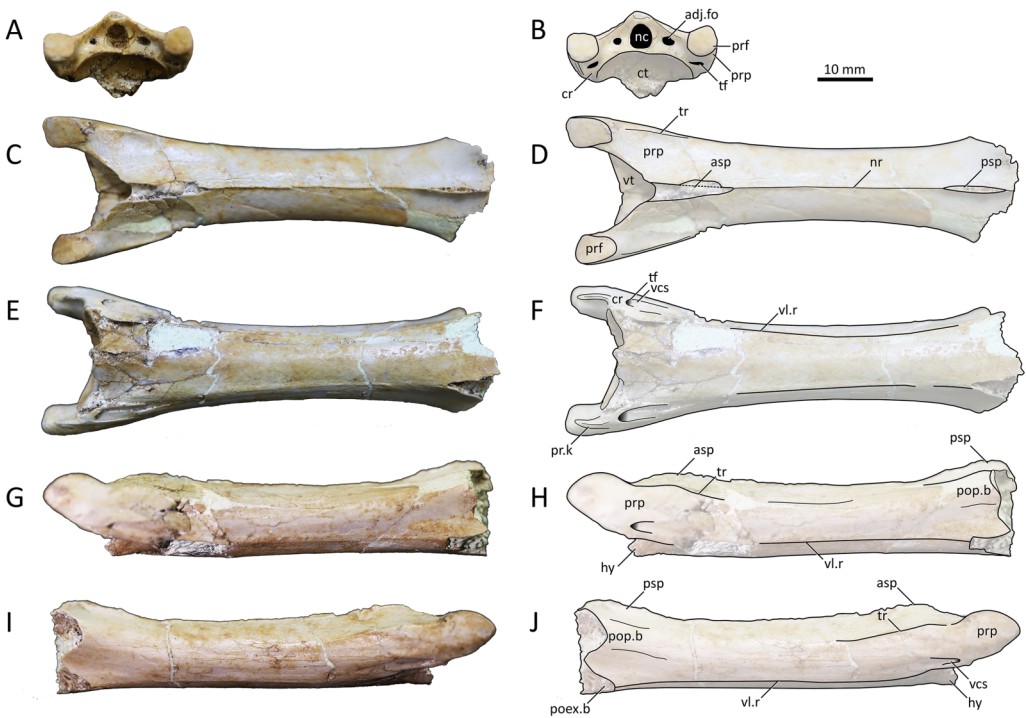

**Figure 5 MPC–Nd 100/303, cervical VI of the holotype of *Tsogtopteryx mongoliensis*.** (A) Anterior view; and (B) schematic drawing; and (C) dorsal view; and (D) schematic drawing; (E) ventral view; and (F) schematic drawing; (G) left lateral view; and (H) schematic drawing; right lateral view; and (J) schematic drawing. Abbreviations: adj.fo, adjacent foramen; asp, anterior spinous process; ct, cotyle; cr, cervical rib; hy, hypapophysis; nc, neural canal; nr, neural ridge; poex.b, postexapophyseal base; pop.b, post-zygapophyseal pedicle base; prf, prezygapophyseal facet; pr.k, prezygapophyseal keel; prp, pre-zygapophyseal pedicle; psp, posterior spinous process; tf, transverse foramen; tr, transverse ridge; vl.r, ventrolateral ridge; vt, vestibule; vcs, vertebrocostal sulcus. Scale bar = 10 mm.

dorsally reflected condition seen in the Burkhant azhdarchid (see above), *Nipponopterus mifunensis* (*Zhou et al., 2024*), *Quetzalcoatlus lawsoni*, and *Arambourgiania philadelphiae* (*Andres & Langston, 2021*). The centrum exhibits a pair of ventrolateral longitudinal ridges that extend, on each side, from the base of the prezygapophyseal pedicle to the base of the postzygapophyseal pedicle (Figs. 5E–5J). This feature is unique to the Bayshin Tsav azhdarchid, being absent in any other known azhdarchid specimen.

The anterior neural canal opening is piriform (Figs. 5A, 5B), with an acuminate dorsal apex. It is bordered by a pair of adjacent foramina, which are level with the ventral half of the neural canal opening. The prezygapophyseal articular facets are piriform in shape, with posteriorly acuminated margins. The prezygapophyseal pedicle ventral surface lacks the large tubercle present in MPC–Nd 100/302b, and instead exhibits an elongated, sharp keel (Figs. 5E, 5F). This feature is unique to *Tsogtopteryx mongoliensis*. A superficially similar structure is present in *Azhdarcho lancicollis*, but in this form the prezygapophyseal pedicle ventral surface exhibits a blunt ridge rather than a sharp keel (*Averianov, 2010*). The vestigial cervical rib is entirely fused to the vertebrae, enclosing a transverse foramen (Figs. 5A, 5B).

MPC–Nd 100/303 has been tentatively interpreted as a cervical IV by *Averianov (2014)*, what was later followed by *Andres & Langston (2021)*. However, as noted by *Andres & Langston (2021)*, azhdarchid fourth cervicals tend to exhibit a posteriorly deflected centrum, a feature that is absent in MPC–Nd 100/303. The discrete lateral constriction and presence of a large notch in the interprezygapophyseal ridge, as found in the sixth cervicals of *Quetzalcoatlus lawsoni* and *Wellnhopterus brevirostris* (*Andres & Langston, 2021*), lead us to reinterpret MPC–Nd 100/303 as a cervical VI.

**Ontogenetic assessment.** Specimen MPC–Nd 100/303 exhibits a fairly smooth, dense bone surface texture, similar to MPC-Nd 100/302 (see above) and other azhdarchid specimens regarded as osteologically mature, such as the Pui azhdarchid (*Vremir et al., 2015*) and *Phosphatodraco mauritanicus* FSAC-OB 12 (*Longrich et al., 2018*). The bone surface does not exhibit, anywhere, the "grained texture" that is typical of young pterosaur specimens (*Bennett, 1993*), which can be seen in the cervical vertebrae of the holotypes of *Cryodrakon boreas* (*Hone, Habib & Therrien, 2019*) and *Albadraco tharmisensis* (*Solomon et al., 2020*). In addition, the specimen exhibits (vestigial) cervical ribs fully fused to the vertebra (enclosing the transverse foramen), which is also characteristic of osteologically mature specimens (*Longrich et al., 2018*; *Andres & Langston, 2021*). Therefore, MPC–Nd 100/303 can be regarded as an osteologically mature individual, despite its small size.

## Phylogenetic analysis

Our phylogenetic analysis produced 27 most parsimonious trees, with 2,177 steps, ensemble consistency index of 0.356 and ensemble retention index of 0.788 (Fig. 6). Similar to previous analyses (*Andres, 2021*), azhdarchids are characterized by three synapomorphies: character 321(2), mesocervical vertebrae neural spines vestigial (bifid; composed of anterior and posterior neural processes, connected by a neural ridge); 348(3), mid-cervical vertebrae extremely elongated (maximum length/width ratio over 5); and 482(3), wing digit phalanges 2 and 3 bearing a ventral keel (ambiguous; unknown in alanqids). Of note, the bifid vestigial neural spines of azhdarchids set them apart from non-azhdarchid azhdarchiforms (see *Andres, 2021*), which exhibit low/reduced but not bifid/vestigial neural spines, as seen in *Montanazhdarcho minor* (see *Andres, 2021*) and alanqids (see *Williams et al., 2021*).

The earliest diverging azhdarchid branch is represented by Phosphatodraconia (Table 1; Fig. 6), supported by character 328(1), absence of adjacent foramina on the mesocervical vertebrae (unknown in *Wellnhopterus*); and 337(1), centrum lateral margins subparallel (ambiguous; unknown in *Aralazhdarcho*). The remaining azhdarchids are joined in a clade, to the exclusion of phosphatodraconians, on the basis of: character 14(0), nasoantorbital fenestra under half of skull length; character 207(1), downcurved mandibular rami; and character 349(1), CVI longer than CIV. The newly defined Quetzalcoatlida is supported by four synapomorphies: character 39(1), jaw tomial edges reduced/rounded; character 184(0), ventralized posterior palate, forming a suspensorium; character 327(1), mesocervical neural canal opening higher-than-wide (in mature forms; presumably confluent neural canal opening and accessory foramen); and character 522(0),

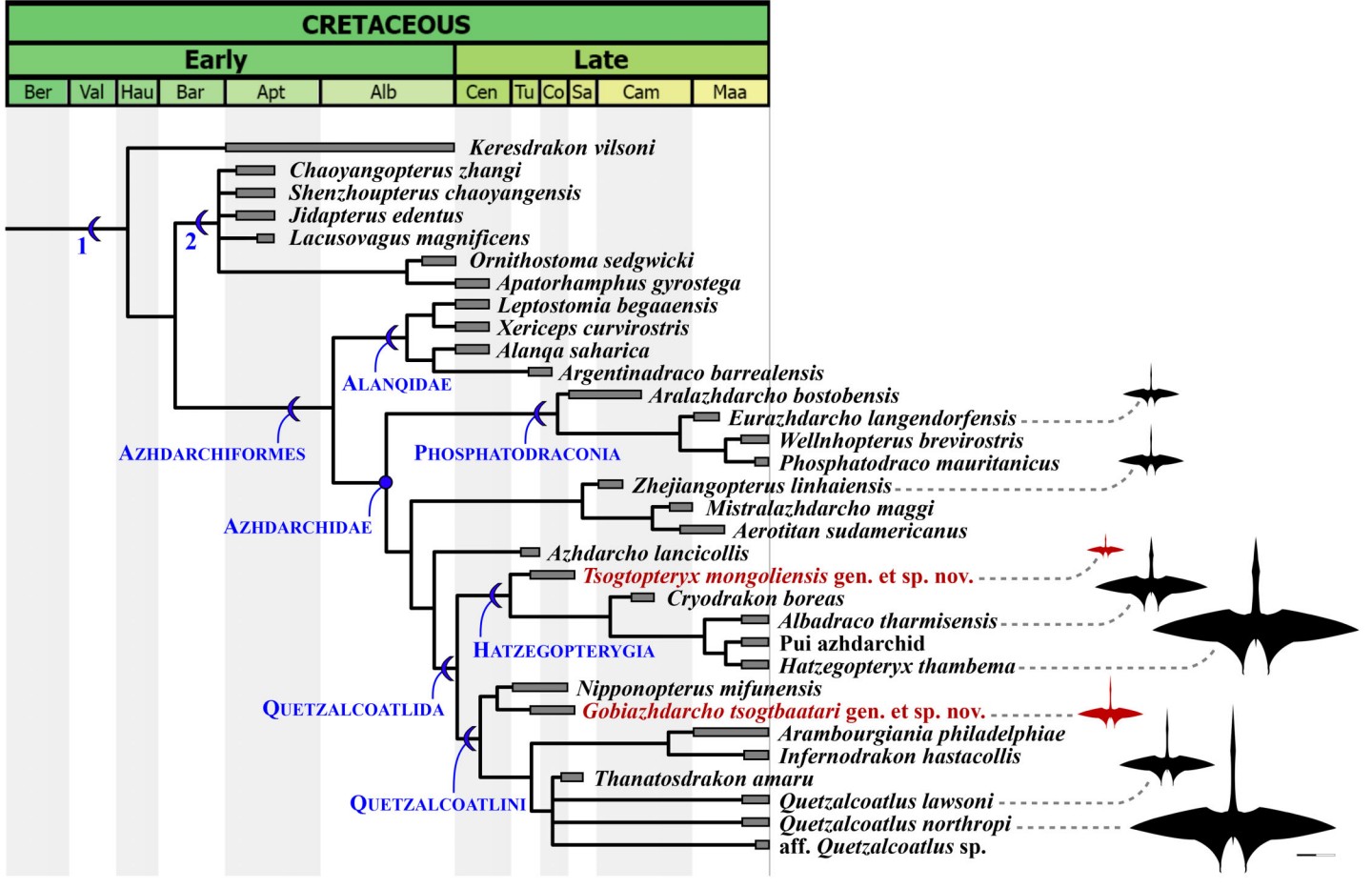

**Figure 6 Time-scaled strict consensus phylogenetic tree.** Partial tree focused on the Azhdarchomorpha (the remaining of the tree is available in the File S3). 1, Azhdarchomorpha; 2, Chaoyangopteridae. Azhdarchid silhouettes modified from the *Quetzalcoatlus* silhouette by Mark P. Witton & Darren Naish, distributed by PhyloPic (phylopic.org/) under a CCBY 3.0 license (https://creativecommons.org/licenses/by/3.0/).

femur shaft strongly bowed (ambiguous; unknown in *Azhdarcho*). See the Discussion for further comments.

Hatzegopterygia is supported by character 323(1), anterior spinous process larger than the posterior one. This character supports placement of *Tsogtopteryx mongoliensis* within the clade. The clade containing the remaining hatzegopterygians, to the exclusion of *Tsogtopteryx mongoliensis*, is supported by character 348(2), mesocervical vertebrae only moderately elongated (length/width ratio under 5). This last feature can be seen in cf. *Hatzegopteryx thambema* (*Vremir, 2010*; *Naish & Witton, 2017*), the Pui azhdarchid (*Vremir et al., 2015*), *Albadraco tharmisensis* (*Solomon et al., 2020*), and *Cryodrakon boreas* (*Hone, Habib & Therrien, 2019*).

Quetzalcoatlini is supported by the following two synapomorphies: character 335(1), mesocervical vertebrae transverse ridges dorsally reflected; and 343(1), mesocervical vertebrae with lateral excavation between postzygapophysis and postexapophysis. These two characters support placement of *Gobiazhdarcho tsogtbaatari* within this group. This

species lies as the sister-taxon of *Nipponopterus mifunensis* (as previously recovered by *Zhou et al. (2024)*), with which it shares character 334(1), dorsally keeled epipophyses; and 342(1), reduced postexapophyses. The clade containing all remaining quetzalcoatlinins is supported by character 344(1), mesocervical vertebrae with a well-developed ventral fossa between the postexapophyses (delineated by an anterior rim).

## DISCUSSION

### The Bayanshiree Formation azhdarchids

According to the present interpretation, the holotypes of *Gobiazhdarcho tsogtbaatari* and *Tsogtopteryx mongoliensis* can be directly compared on the basis of the morphology of cervical VI, which is the only element with preserved overlap between the two specimens. Most importantly, *Tsogtopteryx mongoliensis* exhibits, autapomorphically, a pair of ventrolateral ridges running along the centrum of CVI, which are absent in *Gobiazhdarcho tsogtbaatari*. Furthermore, *Gobiazhdarcho tsogtbaatari* exhibits dorsally reflected transverse ridges and a well-developed posterior lateral fossa (between the postzygapophyses and postexapophyses), both of which are shared with other quetzalcoatlinins but are absent in *Tsogtopteryx mongoliensis*.

It is important to note that, according to the present results, the Burkhant and Bayshin Tsav azhdarchids nest within distinct azhdarchid lineages. *Gobiazhdarcho tsogtbaatari* can be clearly assigned to the Quetzalcoatlini, particularly due to the presence of dorsally reflected transverse ridges and a well-developed posterior lateral fossa (as seen in *Quetzalcoatlus lawsoni* and *Arambourgiania philadelphiae*). In contrast, *Tsogtopteryx mongoliensis* exhibits affinities with Hatzegopterygia instead (particularly due to the anterior spinous process being larger than the posterior one), reinforcing the distinctiveness between the two taxa. At a late-Turonian–Santonian age, *Tsogtopteryx mongoliensis* partially fills a temporal gap within Hatzegopterygia, as the hatzegopterygian lineage was previously restricted to the Campanian–Maastrichtian (*Zhou et al., 2024*), even though quetzalcoatlinins extend from the Turonian–Coniacian to the Maastrichtian (*Zhou et al., 2024*).

Regarding body size, estimates for fragmentary fossil specimens are always uncertain, especially when it comes to groups with a great diversity of skeletal proportions such as azhdarchids (*Cai & Wei, 1994*; *Naish & Witton, 2017*; *Andres & Langston, 2021*). Nonetheless, based on anterior and posterior widths at the zygapophyses as compared to more complete skeletons (such as those of *Q. lawsoni*, *C. boreas*, *M. maggi*, and *Z. linhaiensis*), we herein estimate *Gobiazhdarcho tsogtbaatari* as a medium pterosaur (3.0–3.5 m), and *Tsogtopteryx mongoliensis* as a small one (~1.6–1.9 m wingspan) (see File S3 for further details). It is interesting to note that *Tsogtopteryx mongoliensis* represents one of the smallest known azhdarchid species so far, only behind the ~1.6 meter-wingspan Hornby ?azhdarchid, from the Campanian of Canada (*Martin-Silverstone et al., 2016*).

The Bayanshiree Formation reinforces the reoccurring (though not universal) pattern of multiple, variably-sized azhdarchid species being present in a same deposit (Fig. 7). This pattern has been explored in detail before, based on the co-occurrence of the medium (3 m wingspan) *Wellnhopterus brevirostris*, the large (5 m) *Quetzalcoatlus lawsoni*, and the giant

(10 m) *Quetzalcoatlus northropi* in the Javelina Fm. (*Andres & Langston, 2021*). Similarly, the medium (3 m) Pui azhdarchid, the large (>5 m) *Albadraco tharmisensis*, and the giant (10 m) *Hatzegopteryx thambema* can all be found in the late Maastrichtian of Haţeg Island (*Vremir et al., 2015*; *Solomon et al., 2020*); while the medium (3 m) *Eurazhdarcho langendorfensis* and the giant (~10 m) cf. *Hatzegopteryx* sp. (LPB R2347) both come from the early Maastrichtian of Haţeg Island (*Vremir et al., 2013*, 2018). In the late Maastrichtian of the Ouled Abdoun Basin, at least three azhdarchid species can also be found: the medium (~3 m) aff. *Quetzalcoatlus* sp., the large (4–5 m) *Phosphatodraco mauritanicus*, and the giant (~9 m) cf. *Arambourgiania philadelphiae* (*Longrich et al., 2018*).

### Comments on azhdarchid intrarelationships

According to the present phylogenetic hypothesis, the Azhdarchidae can be subdivided into four main lineages: Phosphatodraconia, a *Zhejiangopterus*-clade, *Azhdarcho*, and Quetzalcoatlida (Fig. 6). A clade (*i.e.*, Quetzalcoatlida) that includes only *Quetzalcoatlus*, *Arambourgiania*, *Hatzegopteryx*, and their closest relatives, to the exclusion of *Azhdarcho lancicollis*, *Zhejiangopterus linhaiensis*, and phosphatodraconians, seems well established (*Andres, Clark & Xu, 2014*; *Andres, 2021*; *Pêgas et al., 2021*, *2023*; *Zhou et al., 2024*). Exactly what further taxa belong within this subclade, or not, remain slightly contentious (*e.g.*, *Longrich et al., 2018*; *Andres, 2021*; *Pêgas, 2024*). Nevertheless, the Quetzalcoatlida as herein proposed is still comparatively stable across distinct phylogenetic proposals; applying to roughly similar clades (regarding composition and diagnosis) under both the present reference phylogeny and alternative ones (*e.g.*, *Andres, 2021*). This is in contrast with the branch-based clade Quetzalcoatlinae *sensu Andres (2021)*, which is comparatively much more unstable in composition (see *Andres, 2021*; *Zhou et al., 2024*). The Quetzalcoatlida can be characterized by four synapomorphies: jaw tomial edges reduced; ventralized pterygoid, forming a suspensorium; presence of a higher-than-wide neural canal opening (in mature forms); and femur shaft strongly bowed (see Phylogenetic Analysis).

The typical higher-than-wide neural canal opening of quetzalcoatlidans is postulated to derive from a confluence between the neural canal opening and the accessory foramen (see *Andres & Langston, 2021*; *Zhou et al., 2024*). This is evidenced from the clithridiate ("keyhole-shaped") neural canals found in *Arambourgiania philadelphiae* (*Martill et al., 1998*), cf. *Hatzegopteryx thambema* (see *Naish & Witton, 2017*), the Pui azhdarchid (*Vremir et al., 2015*), *Quetzalcoatlus lawsoni* (see *Andres & Langston, 2021*), *Gobiazhdarcho tsogtbaatari* (present work), and *Nipponopterus mifunensis* (*Zhou et al., 2024*). Non-quetzalcoatlidan azhdarchids exhibit circular neural canal openings; whether an accessory foramen is present, as in *Azhdarcho lancicollis* (*Nessov, 1984*; *Averianov, 2010*) and *Phosphatodraco mauritanicus* (*Longrich et al., 2018*), or absent (entirely lost), as in the Bakony azhdarchid specimens MTMGyn/448 and MTMGyn/450 (*Ősi & Weishampel, 2005*) and *Mistralazhdarcho maggi* (*Vullo et al., 2018*).

Within the *Quetzalcoatlus lawsoni* hypodigm, while all neural canals are higher-than-wide, some are clearly clithridiate while others exhibit attenuated/acuminated dorsal

margins and less accentuated waists (*Andres & Langston, 2021*). This also contrasts with the typical subcircular condition, and similar shapes can be found in *Tsogtopteryx mongoliensis* (present work), *Albadraco tharmisensis* (*Solomon et al., 2020*), and *Infernodrakon hastacollis* (*Thomas et al., 2025*). Such variation seems related to serial variation within the cervical series (see *Averianov, 2010*; *Andres & Langston, 2021*).

When present, the confluence between the accessory foramen and the neural canal opening seems to be an ontogenetic feature. As noted by *Nessov (1984)*, some Bissekty azhdarchid specimens seem to exhibit varying levels of incipiently confluent accessory foramina and neural canal openings (suggesting the presence of quetzalcoatlidans in the Bissekty azhdarchid assemblage). In *Cryodrakon boreas*, some indication that such incipient confluence could be ontogenetic in nature can also be found. In the holotype of *Cryodrakon boreas*, which is a juvenile specimen (*Hone, Habib & Therrien, 2019*), the preserved mid-cervical (65 mm between prezygapophyses) exhibits distinct accessory foramen and neural canal openings, separated by a well-developed bony bar of regular shape. By contrast, in the larger specimens TMP 1993.40.11 (mid-cervical 82 mm between prezygapophyses) and TMP 1989.36.254 (mid-cervical 97 mm between prezygapophyses), the accessory foramen and neural canal openings are separated by an incipient (extremely thin) bony bar of irregular shape (see *Hone, Habib & Therrien, 2019*). Interestingly, in these specimens, the accessory foramen is larger than the neural canal opening, unlike the holotype wherein the reverse is true. This pattern suggests that, with ontogeny, the accessory foramen expands while the bony bar that separates it from the neural canal opening is reabsorbed (hence its thin, irregular configuration in the larger specimens), ultimately leading to a possible confluence between the accessory foramen and the neural canal opening (note that none of these specimens is fully mature, as indicated by the lack of cervical rib fusion; see *Hone, Habib & Therrien, 2019*). Therefore, we suggest caution regarding the assessment of the presence, or absence, of this feature in azhdarchid taxa. We suggest it can only be confidently assessed when relatively osteologically mature specimens are available, and reiterate that future studies with more complete ontogenetic series will be needed in order to confirm this.

As herein proposed, the Quetzalcoatlida comprise two main lineages: the Hatzegopterygia and the Quetzalcoatlini. A clade comprising the Transylvanian azhdarchids *Hatzegopteryx thambema* and *Albadraco tharmisensis* had already been recovered by *Pêgas et al. (2021, 2023)*; to this clade, Hatzegopterygia, we now assign also *Cryodrakon boreas*, the Pui azhdarchid, and the new species *Tsogtopteryx mongoliensis*. This new phylogenetic structure extends the stratigraphic range of this lineage, previously restricted to the Maastrichtian (*Pêgas et al., 2021*; *Andres, 2021*) or Campanian–Maastrichtian (*Zhou et al., 2024*), back to the Turonian–Santonian. According to the present phylogeny, hatzegopterygians are united by a distinctive feature on the bifid neural spine of the mesocervical series: the anterior spinous process is larger (longer and wider) than the posterior one, unlike other azhdarchids. This feature does not seem to be influenced by serial variation, as demonstrated by the cervical series of *Quetzalcoatlus lawsoni* and *Wellnhopterus brevirostris* (*Andres & Langston, 2021*).

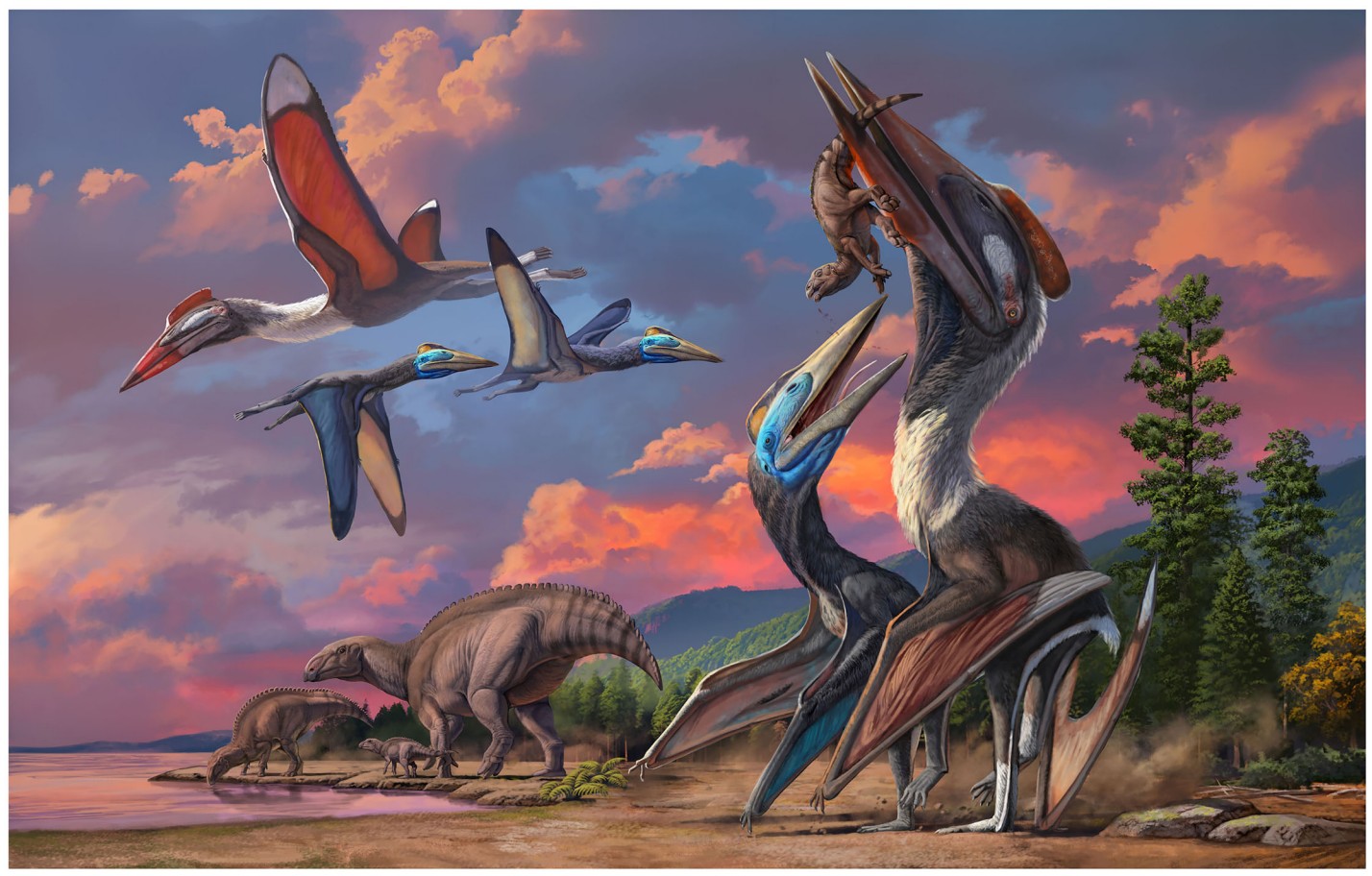

**Figure 7  Life restoration of the Bayanshiree azhdarchids.** The coexistence between *Gobiazhdarcho tsogtbaatari* and *Tsogtopteryx mongoliensis* in the Bayanshiree paleoenvironment, with a group of *Gobihadros mongoliensis* nearby. Artwork by Zhao Chuang.

It is interesting to note that, to the exclusion of the basal form *Tsogtopteryx mongoliensis*, the remaining hatzegopterygians exhibit relatively robust mesocervicals, starkly departing from the typical slender-necked bauplan of other azhdarchids. The existence of a "robust-necked" azhdarchid bauplan was first indicated by the discoveries of cf. *Hatzegopteryx* EME 315 (*Vremir, 2010*) and the Pui azhdarchid (*Vremir et al., 2015*), and later discussed in detail by *Naish & Witton (2017)*. These discoveries indicate that these forms developed secondarily shortened necks relative to other azhdarchids, what implicates a distinct neck biomechanical performance and, in all likelihood, a distinct ecology (*Naish & Witton, 2017*). *Hatzegopteryx*, the Pui azhdarchid, and *Albadraco* had already been recognized as "robust-necked" azhdarchids before (*Vremir et al., 2015*; *Naish & Witton, 2017*; *Solomon et al., 2020*), and *Cryodrakon boreas* seems to represent at least an intermediate morphology regarding robustness (see *Hone, Habib & Therrien, 2019*).

In turn, Quetzalcoatlini comprises all quetzalcoatlidans closer to *Quetzalcoatlus northropi* than to *Hatzegopteryx thambema* (Table 1). At the base of this clade lie *Gobiazhdarcho tsogtbaatari* and *Nipponopterus mifunensis*, which share with other

quetzalcoatlinins the following two features: dorsally reflected transverse ridges, and a well-developed lateral fossa between the postzygapophyses and postexapophyses (*Zhou et al., 2024*). The dorsally reflected ridges can be seen in remains of *Arambourgiania philadelphiae* (*Frey & Martill, 1996*), *Infernodrakon hastacollis* (*Thomas et al., 2025*), *Quetzalcoatlus lawsoni* (*Andres & Langston, 2021*), and aff. *Quetzalcoatlus* sp. (*Longrich et al., 2018*), as well as *Gobiazhdarcho tsogtbaatari* and *Nipponopterus mifunensis* (*Zhou et al., 2024*). Regarding the lateral fossa between the postzygapophyses and postexapophyses, this feature is bordered ventrally by a sharp flange that protrudes from the lateral surface of the postexapophyseal peduncle. This latter feature can be particularly well-seen in remains of *Infernodrakon hastacollis* (*Thomas et al., 2025*), *Quetzalcoatlus lawsoni*, and cf. *Quetzalcoatlus northropi* (*Andres & Langston, 2021*), aside from *Gobiazhdarcho tsogtbaatari*. This feature is also present in *Wellnhopterus brevirostris* (see *Andres & Langston, 2021*), although this is recovered here as a homoplasy. Interestingly, this lateral fossa bears a pneumatic opening in remains of cf. *Arambourgiania philadelphiae* (*Martill & Moser, 2018*) and *Infernodrakon hastacollis* (*Thomas et al., 2025*). To the exception of *Gobiazhdarcho tsogtbaatari*, the remaining quetzalcoatlinins further share a well-defined ventral fossa between the postexapophyses (delineated by an anterior rim), as seen in remains of cf. *Quetzalcoatlus northropi*, *Quetzalcoatlus lawsoni* (*Andres & Langston, 2021*), aff. *Quetzalcoatlus* sp. (*Longrich et al., 2018*), and *Arambourgiania philadelphiae* (*Martill & Moser, 2018*).

## CONCLUSIONS

The present anatomical reassessment of the Bayanshiree Fm. azhdarchids reveals several features that distinguish the Burkhant and Bayshin Tsav specimens from each other, as well as from other azhdarchids. These specimens are accordingly recognized here as new species: *Gobiazhdarcho tsogtbaatari*, the Burkhant azhdarchid; and *Tsogtopteryx mongoliensis*, the Bayshin Tsav azhdarchid. These are identified, respectively, as members of the newly recognized clades Quetzalcoatlini and Hatzegopterygia. These forms reiterate the general pattern of a single deposit yielding multiple azhdarchid species of distinct body sizes, with *Tsogtopteryx mongoliensis* being notably small for an azhdarchid (with a wingspan of ~1.6–1.9 m) whereas *Gobiazhdarcho tsogtbaatari* represents a medium (3.0–3.5 m) pterosaur. Apart from enhancing our understanding of the diversity of the Bayanshiree Fm., the recognition of these two new species also fills in important temporal gaps in the evolutionary history of azhdarchids.

## ACKNOWLEDGEMENTS

We are deeply grateful to Khishigjav Tsogtbaatar (MPC) for encouraging the development of the present research. We also acknowledge Mr. Ken Hayashibara (president of the Hayashibara Company Limited, Okayama, Japan) and all participants of the *Hayashibara Museum of Natural Sciences-Mongolian Paleontological Center Joint Paleontological Expeditions* for their contributions to Mongolian paleontology. We further thank the Willi Hennig Society for making TNT freely available, Zhao Chuang for kindly providing his

superb artwork, and reviewers Henry Thomas and Leo Ortiz David for their constructive remarks.

## INSTITUTIONAL ABBREVIATIONS

| | |
|---|---|
| **EME** | Transylvanian Museum Society, Cluj-Napoca, Romania |
| **FSAC** | Faculté des Sciences Aïn-Chock, Université Hassan II, Casablanca, Morocco |
| **LPB** | Laboratory of Fossil Vertebrates, Faculty of Geology and Geophysics, University of Bucharest, Bucharest, Romania |
| **MPC** | Mongolian Paleontological Center, Mongolian Academy of Sciences, Ulaanbaatar, Mongolia |
| **MTM** | Magyar Természettudományi Múzeum, Budapest, Hungary |
| **SNSB-BSPG** | Staatliche Naturwissenschaftliche Sammlungen Bayerns/Bayerische Staatssammlung für Palaontologie und Geologie, Munich, Germany |
| **TMM** | Texas Vertebrate Paleontology Collections, The University of Texas at Austin, Austin, Texas |
| **TMP** | Royal Tyrrell Museum of Palaeontology, Drumheller, Alberta, Canada |
| **ZIN** | Zoological Institute of the Russian Academy of Sciences, St. Petersburg, Russia |
| **ZMNH** | Zhejiang Museum of Natural History, Hangzhou, China |

### Funding
This work was supported by the Fundação de Amparo à Pesquisa do Estado de São Paulo (No. 2023/11296-0). The Shihezi University High-level Talents Research Startup Project (RCZK202597) paid the APC charge for this article. The funders had no role in study design, data collection and analysis, decision to publish, or preparation of the manuscript.

### Grant Disclosures
The following grant information was disclosed by the authors:
Fundação de Amparo à Pesquisa do Estado de São Paulo: 2023/11296-0.
Shihezi University High-level Talents Research Startup Project: RCZK202597.

### Competing Interests
The authors declare that they have no competing interests.

### Author Contributions
- R. V. Pêgas conceived and designed the experiments, performed the experiments, analyzed the data, prepared figures and/or tables, authored or reviewed drafts of the article, and approved the final draft.

- Xuanyu Zhou conceived and designed the experiments, performed the experiments, analyzed the data, prepared figures and/or tables, authored or reviewed drafts of the article, and approved the final draft.
- Yoshitsugu Kobayashi conceived and designed the experiments, authored or reviewed drafts of the article, and approved the final draft.

## Data Availability

The phylogenetic matrix is available in the Supplemental File.

## New Species Registration

The following information was supplied regarding the registration of a newly described species:

Publication LSID: urn:lsid:zoobank.org:pub:72240BB4-B98C-40B4-ADBC-A1DEDE9E06DA.

*Gobiazhdarcho* genus LSID: urn:lsid:zoobank.org:act:A675EB7D-3502-4F99-8446-8AE4918AC60A.

*Gobiazhdarcho tsogtbaatari* species LSID: urn:lsid:zoobank.org:act:59F3DABA-6E84-4DE2-8948-F5F250E2E910.

*Tsogtopteryx* genus LSID: urn:lsid:zoobank.org:act:A724E4E3-A6EA-415E-9A12-0B8F0636FEF0.

*Tsogtopteryx mongoliensis* species LSID: urn:lsid:zoobank.org:act:3F218EE5-2CB8-469D-A252-C5194C7C6911.

## Supplemental Information

Supplemental information for this article can be found online at http://dx.doi.org/10.7717/peerj.19711#supplemental-information.

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
