# Peer review of "Azhdarchid pterosaur diversity in the Bayanshiree Formation, Upper Cretaceous of the Gobi Desert, Mongolia"

_PeerJ, doi:10.7717/peerj.19711_

## Round 0.1 · original submission · Minor Revisions

Most critically for revision, both reviewers note that the diagnoses should be revised, to separate autapomorphies from unique combinations of synapomorphies. This should be completed in the revised manuscript.
The reviewers identify a number of other smaller suggestions; please address these as appropriate in your revision and response document.

·

Basic reporting

The manuscript is well-organized and logically structured for a taxonomic work. The language is direct and accessible, but at a couple points the phrasing comes off as overly casual. I have some grammatical and phrasing suggestions to rectify this and increase clarity in the text, which the authors may consider as they see fit:

Line 28: Add a specifier, such as “in the region”, following “abundance of dinosaur fossils”
Line 55: “counts with” → “includes”
Line 66: “what results” → “resulting”
Lines 87-88: “comprising” used thrice in short succession; I recommend changing it up a bit
Line 96: “This topic” – please specify that the topic is azhdarchid diversity/systematics
Lines 112, 441, 710, and Supplemental Text: “notoriously” is not the most fitting word to use in these sentences (as the small size of the Bayshin Tsav azhdarchid is first commented on in this study, it doesn’t exactly have much notoriety). Consider substituting with words such as “significantly”, “particularly”, or “notably”.
Line 120: “on the literature” → “in the literature”
Lines 197, 593, 704, 712: I recommend consistently spelling out “Formation” instead of abbreviating it as “Fm.” in the main body of the text.
Line 409: “the” before Nipponopterus can be removed
Line 481: Please specify “lateral margins of the centrum”
Line 511: “what is” → “which is”
Line 562: “long” → “along”
Line 571: “These features are absent in Tsogtopteryx mongoliensis” is redundant with the previous paragraph and can be deleted.
Line 572: “affinities to” → “affinities with”
Lines 579-580: “filled with uncertainty” can be shortened to “uncertain”
Lines 589: “It is interesting to note that” can be removed from the beginning of this sentence. The same phrasing is used verbatim in the previous sentence, and removing it would strengthen the leading statement of this paragraph.
Line 597: I would replace “stem from” with another phrase, such as “occur in”
Line 607: “their respective most closely related forms” can be condensed to “their closet relatives”
Lines 675-676: “secondarily shorter” can be changed to “secondarily shortened”
Line 676: “implicates in” → “implicates”
Line 680: “intermediate morphology” does not need to be in quotation marks
Line 689: “aside from” → “as well as”
Figure 6 caption: “remaining” → “remainder”

The manuscript is well-sourced and does not omit any relevant literature. The figures are clear and informative. Figure 7 is gorgeous; give Zhao Chuang my regards! My only comments regard Figure 6. This figure uses closed blue circles to denote minimum-clades on the phylogeny, and open brackets to denote maximum-clades. I appreciate this visual indicator. However, the new clades Hatzegopterygia and Quetzalcoatlini are represented by closed circles. Table 1 defines both of these as maximum-clades, so the symbols should be corrected to open brackets. Also, the new clade Phosphatodraconia is not labeled on the tree, and I see no reason why it shouldn’t be. Figure 6 is cited alongside Table 1 when Phosphatodraconia is introduced in the text (lines 527-528), so it should be labeled on the tree.

Experimental design

The aim of this study is to redescribe, taxonomically reassess, and determine the phylogenetic position of two azhdarchid specimens from the Bayanshiree Formation in light of a greater understanding of azhdarchid morphology and diversity. To that end, it does strong work. The taxonomic practices in this manuscript are robust and meet ICZN and PhyloCode standards, as well as journal requirements for new species. I commend the authors’ transparency regarding anatomical terminology and species circumscription. The phylogenetic dataset is comprehensive, and the phylogenetic methods are straightforward and sufficient for me to replicate the analysis and recover the same tree. I also appreciate character-by-character description of the makeup of Azhdarchidae in the phylogenetic results section.

Validity of the findings

The authors provide a strong argument that Gobiazhdarcho represents a unique taxon and a close relative of Nipponopterus, in spite of the fragmentary nature of the holotype. I recommend that the diagnosis state which characters are autapomorphic to G. tsogtbaatari and which are diagnostic via combination. Reduced postexapophyses and keeled dorsal epipophyses are also present in Nipponopterus; these characters can be included in a combinatorial diagnosis, but cannot be considered autapomorphies of Gobiazhdarcho.

The description of Tsogtopteryx would benefit from additional comparisons to further demonstrate that it represents a unique taxon. As currently written, for example, it is unclear whether the ventral keels on the prezygapophyseal pedicles are autapomorphic. Based on the figures included in Nesov (1984) and Averianov (2010), it looks like the holotype of Azhdarcho lancicollis (a cervical V) might have similar prezygapophyseal keels to MPC-Nd 100/303. It would behoove the authors to elaborate further on why MPC-Nd 100/303 represents a unique azhdarchid morphotype, perhaps by highlighting that this specimen displays a unique combination of characters alongside the autapomorphic longitudinal ridges. This would strengthen the diagnosis of the Bayshin Tsav taxon and its inferred phylogenetic position.

The insights in the Discussion section are logical extensions of the systematic work in this paper, adding to our understanding of azhdarchid systematics, and the Conclusions summarize the results and implications of this work well. I would recommend that the Conclusions clarify that Quetzalcoatlini and Hatzegopterygia are new clades. The supplementary material all checks out to me, and I have no major comments there.

Additional comments

Line 76: Please include age of the Öösh Formation for consistency
Line 80: Please include age of the Tugrikin Shireh beds for consistency
Line 113: Spell out “eight”
Lines 233-235: ZIN should be listed before ZMNH
Line 259: As acknowledged later on in the manuscript (lines 351, 357-358) the azhdarchid cervical III still has a relatively tall and non-bifid neural spine. Therefore, this line should probably be amended to “Mid-cervical vertebrae neural spines vestigial”, so that the statement isn’t applied to cervical III too.
Lines 282-284: It may be worth noting that the unrestricted emendation of Azhdarchidae does not change clade content if retroactively applied to phylogenies based on Andres and Myers (2013) and its derivatives, since the two new internal specifiers are always found within the (Azhdarcho+Quetzalcoatlus) clade.
Lines 327-344: I recognize that the species circumscription of Aralazhdarcho bostobensis is restricted to the holotype mesocervical, but would it be worth mentioning the atlantoaxis (ZIN PH 44/43) referred to it by Averianov (2007)? Systematic variation in the azhdarchid atlantoaxis is not as well understood as the mesocervicals, and that specimen has a lateral pneumatic foramen like Mistralazhdarcho, so it may warrant at least an acknowledgement.
Line 383: Typo in Averianov, 2010 citation
Lines 409 and 629: Please cite specimen numbers instead of “the Bakony azhdarchid”. There are multiple cervical vertebrae described in Ősi et al. (2005) and it is uncertain whether these are all conspecific.
Line 489: Missing “azhdarchid” after “Bayshin Tsav”
Line 576: Cryodrakon is known from the upper Campanian and is also recovered within Hatzegopterygia in this study and in Zhou et al. (2024). So, apart from Tsogtopteryx, Hatzegopterygia is restricted to the Campanian-Maastrichtian, not the Maastrichtian.
Line 690: Typo in “postzygapophyses”

·

Basic reporting

The work satisfies all the requested submission standards. It is sufficiently well written, with correct use of technical vocabulary, has the citations expected in this type of work and presents a correct structure.

Experimental design

The research is correct in its methodology and development. It presents a high technical standard and develops all the sections that should be considered in a work that includes anatomical and phylogenetic analysis and comparisons.

Validity of the findings

no comments

Additional comments

I have reviewed and analyzed the paper entitled “Azhdarchid pterosaur diversity in the Bayanshiree Formation, Upper Cretaceous of the Gobi Desert, Mongolia” at the request of the editorial board of the prestigious journal PeerJ.
I would like to start by congratulating the authors for the very good work done, where in light of the advancement in the current knowledge of the different pterosaur lineages, the previously recovered materials take on a greater importance. I emphasize that the work has the structure and quality to be published. It has detailed anatomical analysis, deep comparisons with key taxa and updated bibliography. In addition, a phylogenetic analysis is performed with very interesting results, the establishment of new clades and two genera and species of azhdarchid pterosaurs.

The paper performs a review of pterosaur remains previously found in two nearby localities located in southern Mongolia in the Bayanshiree Formation. These remains are part of the very abundant vertebrate record of the country, which contrasts with the very scarce record of pterosaurs. In this context, all the information that can be extracted from this fossil material is important to understand one aspect of the faunas that inhabited the Cretaceous.

The work clearly shows that both specimens can be confidently assigned to the Azhdarchidae clade and that they present sufficient characteristics to be considered as two different species. The azhdarchids of Burkhant and Bayshin Tsav form an association that integrates with others previously defined, where different species occupied the same ecosystem.

I consider that the work requires moderate revisions (major revisions are indicated as corrections should be re-evaluated), due to what is indicated in the pdf. The comments made in a summarized manner concern the definition of the new taxa and the anatomical characteristics highlighted. A modification in the diagnosis of both taxa is required since as they are they do not allow to define them correctly. I suggest separating and highlighting the autapomorphic characters and including sets of synapomorphic characters that in combination allow identifying that these specimens are new genera and species. Likewise, I suggest deepening some comparisons to emphasize punctually that the characters indicated are autapomorphies. Finally, this should be accompanied by the inclusion of photo-drawings in the figures already made where these structures are punctually highlighted so that they can be observed without problems. Since this is a research where the description of two new taxa is the central theme, I consider that these modifications are crucial for the work.

---

## Round 0.2 · accepted · Accept

Thank you for your close attention to the reviewers' comments. I have assessed the revision, and in my opinion you have adequately addressed the concerns with the previous version.

The manuscript is, in my opinion, ready to proceed to publication. I did note one minor grammatical edit, which may be handled directly without my review. Specifically:

p. 15, line 459: "acuminated in shape" - it is sufficient to say "acuminate" (which describes the shape)